# Diurnal Asymmetry in Nonlinear Responses of Canopy Urban Heat Island to Urban Morphology in Beijing during Heat Wave Periods

Tao Shi[1,2,3], Yuanjian Yang[4*], Ping Qi[1], Simone Lolli[5]

[1]School of Mathematics and Computer Science, Tongling University, Tongling, 244000, China
[2]School of Geography and Tourism, Anhui Normal University, Wuhu, 241000, China
[3]Key Laboratory of Transportation Meteorology of China Meteorological Administration, Nanjing Joint Institute for Atmospheric Sciences, Nanjing, 210041, China
[4]State Key Laboratory of Climate System Prediction and Risk Management, School of Atmospheric Physics, Nanjing University of Information Science and Technology, Nanjing 210044, China
[5]CNR-IMAA, Contrada S. Loja, 85050 Tito Scalo (PZ), Italy

*Correspondence to*: Prof. Yuanjian Yang (yyj1985@nuist.edu.cn)

**Abstract.** Currently, the diurnal asymmetric and nonlinear mechanisms by which urban morphology modulates the canopy urban heat island (CUHI) during heat wave (HW) periods remain underexplored. This study aims to fill this gap by focusing on the area within the Fifth Ring Road of Beijing, integrating three complementary methods: XGBoost (to identify key morphological drivers), ENVI-met (to reveal nonlinear regulatory processes), and wind environment analysis (to supplement dynamic modulation). The results show that: (1) HW periods significantly enhance CUHI intensity (CUHII) compared to non-heat wave (NHW) periods, with a 91.3% increase in daytime and 52.7% at night; (2) XGBoost identifies building coverage ratio (BCR) as the core daytime driver of CUHII, while sky view factor (SVF) dominates at night, and both 2D and 3D morphological indicators exert stronger effects during HW periods; (3) ENVI-met simulations reveal nonlinear mechanisms of building height/SVF: daytime thermal environments are co-driven by short-wave radiation shading and ventilation resistance (as SVF decreases), while nighttime environments are dominated by long-wave radiation accumulation by buildings; (4) Wind environment analysis further shows diurnal differences in wind's role: nighttime ventilation corridors mitigate CUHII by 33.91–42.09%, while daytime prevailing winds may exacerbate downstream CUHII via thermal advection. These findings clarify the diurnal asymmetric mechanisms of CUHI and provide scientific support for urban morphological optimization under extreme heat.

## 1 Introduction

The latest assessment report from the Intergovernmental Panel on Climate Change (IPCC) indicates a significant increase in the frequency, intensity, and duration of extreme heat events (IPCC, 2021). The CUHI, the phenomenon of abnormal air temperature from near-surface to roof height, has become a research focus due to its direct impact on outdoor thermal comfort and building energy consumption (Battista et al., 2023; Shi et al., 2024). Notably, in particular, the CUHII is amplified significantly during HW periods. In megacities such as Beijing, Shanghai and Guangzhou, China, the intensity of

canopy heat island increases by 0.8 to 1.2 °C during HW periods (Jiang et al., 2019; Yang et al., 2023), with a marked expansion in diurnal amplitude (Ao et al., 2019; Shi et al., 2024).

Against the backdrop of urban heat island mitigation, deciphering the mechanism by which complex urban morphology drives local thermal environments is of critical scientific significance (Berger et al., 2017; Huang & Wang, 2019; Wu et al., 2022; Guo et al., 2023). Existing studies show that two-dimensional urban morphological indicators (e.g., building area ratio, aggregation index) are key controlling factors for local thermal environments (Henits et al., 2017; Shi et al., 2021). With the widespread use of three-dimensional building data, research confirms that three-dimensional morphological indicators such as building height, SVF exhibit stronger explanatory power for local thermal environments (Shao et al., 2023; Zhang et al., 2023; Ding et al., 2024). Although there are conflicting conclusions regarding the correlation between three-dimensional morphological elements and thermal environments (e.g., SVF showing a positive, negative, or no significant correlation with local temperature) (Huang & Wang, 2019; Li & Hu, 2022), their inclusion in models can significantly enhance the explanatory power of urban heat island intensity (Wu et al., 2022). However, existing studies have not clarified the impacts of 2D/3D morphology on daytime and nighttime CUHII and their driving mechanisms, and systematic analysis during HW periods is even more lacking.

Current research on the nonlinear relationship between urban morphology and local thermal environments focuses primarily on surface thermal environments (Han et al., 2022; Wu et al., 2022; Guo et al., 2023; Gu et al., 2024; Wang et al., 2024; Liu et al., 2025). For example, Gu et al. (2024) found that the enhancement effect of floor area ratio on land surface temperature tends to saturate when floor area ratio exceeds 0.6, and the impact of building height on LST slows when building height exceeds 15 meters. In particular, due to fundamental differences in physical mechanisms between the air temperature of the urban canopy (based on the thermodynamic processes of the canopy air) and the surface temperature (based on the energy balance of surface radiation), these conclusions cannot be applied directly to CUHI research. Traditional statistical models such as Ordinary Least Squares (Wang et al., 2020), Spatial Autocorrelation Model (Fallah Ghalhari and Dadashi Roudbari, 2018), and Geographically Weighted Regression (Gao et al., 2022) have inherent limitations in handling nonlinear relationships (Alonso & Renard, 2020), while machine learning methods – through feature importance analysis and SHAP value interpretation (Lundberg & Lee, 2017) – offer a new technical approach. Furthermore, ENVI-met, a three-dimensional non-hydrostatic model, enables microclimate simulation at 0.5–10 m spatial resolution and 1–10 s temporal resolution by coupling short/long-wave radiation budget processes on building surfaces (Chan & Chau, 2021), providing a powerful tool for fine-scale analysis of regional microclimate mechanisms (Meng et al., 2024; Luo et al., 2024).

Beijing, as a typical fast-developing megacity, exhibits significant spatial heterogeneity in urban morphology due to its polycentric ring development pattern (Jiang et al., 2024), offering an ideal case for studying diurnal differences in CUHII during HW periods. This study integrates ground observation data and high-precision urban morphology data, combining machine learning and numerical simulation methods to systematically explore the contributions of key three-dimensional urban morphological indicators to CUHII during HW periods and the diurnal variations in their nonlinear modulation. The findings will not only provide quantifiable morphological indicators for the management of urban extreme heat risk, but will

also provide scientific information on the diurnal variations of CUHII and their potential causes.

## 2 Data and methodology

### 2.1 Study Area

Beijing megacity is located at the northern end of the North China Plain, featuring a complex terrain: the Yan Mountain and Taihang Mountain with altitudes exceeding 2000 meters adjoin theirnorth and west, the northeast is hilly, the south is plain, and the southeast extends to the Bohai Bay to form a land-sea transition zone. From 1978 to 2022, Beijing's population increased from 8.71 million to 21.84 million, with 41.8% of the permanent population concentrated within the Fifth Ring Road, which accounts for only 4.07% of the city's area, demonstrating significant population agglomeration. This study focuses on the area within the fifth ring road (Figure 1). As central urbanization area of Beijing, the spatial heterogeneity of population density, building distribution, and green space configuration in this region provides a typical scenario for urban thermal environment research. Although the urban green coverage rate increased from 22.3% in 1978 to 49.0% in 2020, the intensity of the heat island still increased at a rate of 0.24°C/year (Ge et al., 2016). Based on the air temperature data during the NHW and HW periods in summer, this study focuses on analyzing the diurnal variations of CUHII and exploring its correlation mechanism with urban morphology.

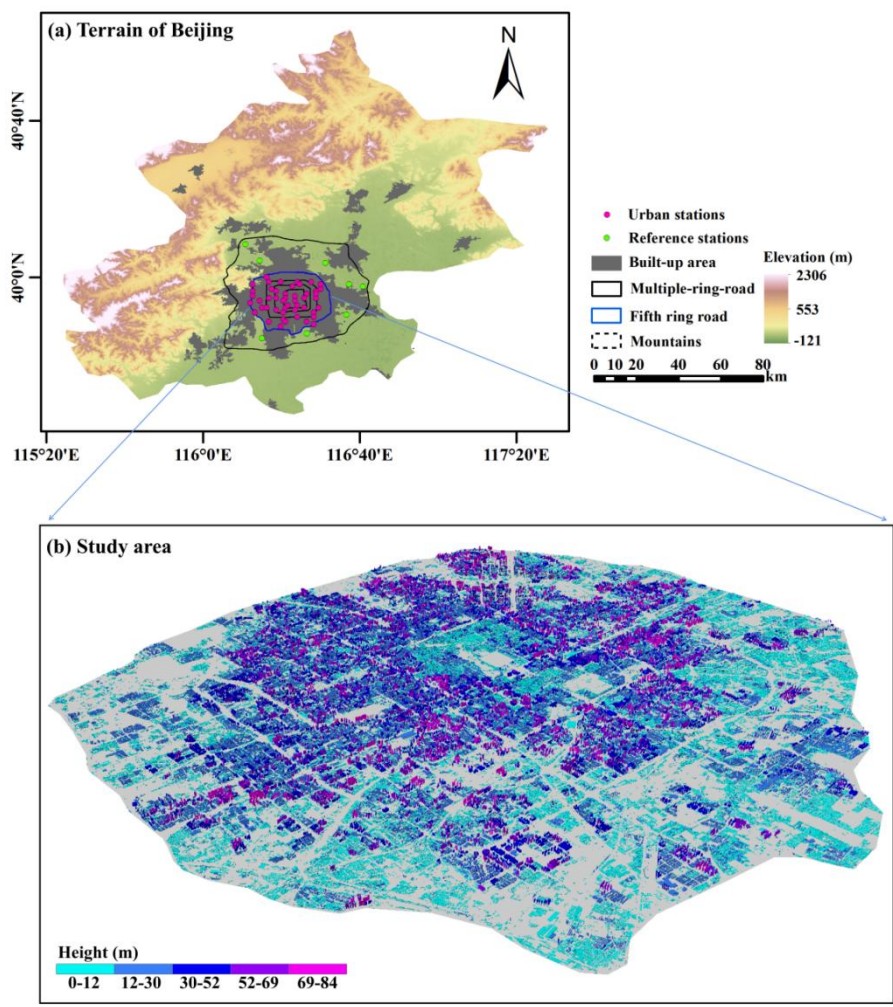

**Figure 1: Overview of the study area. (a) Topography and land use in Beijing, with distribution of urban and reference observation stations in the built-up area of the city. (b) Urban morphological characteristics of the study region.**

## 2.2 Data collection and processing

### 2.2.1 AWS observation data

The hourly observation data from automatic weather stations (AWS) used in this study were obtained from the China Meteorological Data Service Center (http://data.cma.cn/en), including meteorological elements such as near-surface air temperature, wind speed, wind direction, humidity, and precipitation. To ensure data accuracy, we performed quality control on the ground station observation data: referring to previous research methods (Yang et al., 2011; Xu et al., 2013), we

imputed missing values in the observation sequences using the average of synchronous observation data from the five nearest stations around the site and excluded station records with excessive errors. The final AWS observation dataset from

2018-2022 was used to analyze the spatio-temporal distribution characteristics of the near-surface thermal field in the Beijing megacity.

### 2.2.2 Selection and calculation of urban morphology indicators

From a spatial perspective, urban spatial morphology can be divided into urban 2D/3D morphology. At the 2D level, academic circles have systematically explored the association between urban morphology and local thermal environments (Tysa et al., 2019; Yu et al., 2020). For instance, the proportion of building area has a significant warming effect (Wang et al., 2017; Liu et al., 2021), and studies have shown that when the building area is fixed, there is a significant positive correlation between temperature and the building patch index (Shi et al., 2015). In addition to 2D morphology, the regulatory role of 3D urban morphology in thermal environments has attracted much attention in recent years (Yin et al., 2018; Tian et al., 2019; Zhou et al., 2022; Xu et al., 2024; Bansal & Quan, 2024). Although 3D morphology is based on 2D pattern parameters with the addition of height information, its characterization is not limited to height but also includes other features derived from height. Taking the sky view factor (SVF) as an example, this indicator refers to the ratio of the visible sky range to the total visible range at a fixed point on the ground. It is an important parameter for characterizing the geometric characteristics, density, and thermal balance of urban areas, and also a key factor affecting the generation and intensity of the heat island effect (Scarano & Mancini, 2017). Relevant studies have shown that surface temperature in summer is significantly correlated with building height (Cai & Xu, 2017); regulating SVF may serve as a potential means to mitigate the urban local thermal environment in high-density urban areas (Xu et al., 2024). We obtained building data from Baidu Maps (https://map.baidu.com), including building base projection boundaries and total floor information. The building base projection boundaries can be used to characterize the horizontal distribution of urban buildings. We calculate the height of the building by multiplying the number of floors by 3 meters. This method has been verified to have an overall accuracy of 86.78% (Liu et al., 2021), and the conversion results are reliable based on the regular characteristics of the floor heights of urban buildings (Alavipanah et al., 2018). The specific definitions and calculations of the 2D/3D indicators are as follows in Table S1. Finally, we selected a 500m buffer zone (Oke, 2004) and used the six two-dimensional indicators and six three-dimensional indicators to describe the morphological characteristics of buildings.

### 2.3 Method implementation

### 2.3.1 NHW and HW periods classification

Global standards for defining HW events vary significantly due to climatic backgrounds, geographical conditions, and socioeconomic factors. The World Meteorological Organization (WMO) defines a heat wave as three consecutive days with maximum temperatures exceeding 32°C; the National Oceanic and Atmospheric Administration (NOAA) uses a heat wave index that integrates temperature and humidity, issuing alerts when the index exceeds 40.5°C for at least three hours per day for two consecutive days or forecasts reach 46.5°C; the Royal Netherlands Meteorological Institute requires five consecutive days with maximum temperatures over 25°C, including at least three days exceeding 30°C. This study adopts the China Meteorological Administration (CMA) definition of heat waves as three consecutive days with maximum temperatures ≥35°C. Considering that maximum temperatures at urban stations may be influenced by urbanization, we identify heat waves

based on reference station data in this study: In summer, a day is classified as a HW day if more than two reference stations simultaneously meet the CMA heat wave criteria; otherwise, it is a NHW day. This method ensures HWs are recognized as regional extremes rather than local anomalies. A single reference station's high temperatures may stem from microtopography or temporary activities (Perkins et al., 2022), while ≥2 stations confirm spatial consistency, reducing misclassification and aligning with HWs' large-scale pattern (Rajulapati et al., 2022; Xue et al., 2023).

### 2.3.2 CUHII quantification

Academia typically defines CUHII as the temperature difference between urban and reference stations (Yang et al., 2023; Shi et al., 2024). The selectionof reference stations is critical for the calculation of CUHII, adhering to specific criteria: 1) Significantly lower temperatures than urban stations; 2) Location in rural forest-shrub areas more than 50 km from the city center (Yang et al., 2023); 3) Uniform distribution in different urban orientations. Finally, we selected 8 reference stations (green markers in Figure 1), with an average altitude of 39.6 m, 8.8 m lower than the 45 urban stations (red markers in Figure 1). We obtained summer CUHII values for urban stations by calculating temperature differences between urban and reference stations.

### 2.3.3 Machine learning model

Compared to traditional machine learning methods, the XGBoost algorithm demonstrates significant advantages in accuracy, flexibility, anti-overfit capability, and missing value processing (Chen et al., 2023). Its superior performance stems from loss function optimization based on second-order Taylor expansion, multithread parallel computing support, and regularization constraint mechanisms (Chen & Guestrin, 2016). Traditional linear regression models struggle to capture the nonlinear local characteristics between influencing factors and thermal environments, while XGBoost can effectively analyze the nonlinear mechanism between factor changes and local thermal environments (Lin et al., 2024). In this study, we first performed iterative calculations on 7 commonly used hyperparameters (eta, gamma, max_depth, min_child_weight, subsample, colsample_bytree, and nrounds) within a preset hyperparameter tuning space, and selected the optimal hyperparameter combination that minimizes model error using a 5-fold cross-validation method (Yang et al., 2020; Lin et al., 2024). After completing hyperparameter optimization, we randomly split the sample points in the Beijing at a 7:3 ratio to obtain training samples (70%) and validation samples (30%), which were used for training and validating the XGBoost model, respectively. Meanwhile, the coefficient of determination ($R^2$) and root mean square error (RMSE) were chosen as evaluation metrics for simulation accuracy.

Additionally, we introduce the SHAP model in this study to improve interpretability, which quantifies the impact of each morphological parameter on the thermal environment through global and local variable attribution (Hong et al., 2025). SHAP (SHapley Additive exPlanations): This method quantifies each feature's contribution to individual predictions based on Shapley values from game theory (Park et al., 2023). For each sample, SHAP values decompose the prediction into feature-specific contributions, with positive/negative values indicating promotion/inhibition of CUHII. Partial dependency plots (PDP) are a common explainable machine learning technique that reveals the marginal effect of a target feature (e.g.,

urban morphological indicators) on prediction outcomes (CUHII) by holding other features at their average levels or marginalizing their effects (Friedman, 2001; Bansal & Quan, 2024). Specifically, PDP illustrates the average trend of change in CUHII as a single indicator (or a combination of two indicators) varies, while other indicators remain stable—thereby isolating the independent impact of the target indicator. By leveraging PDP to visualize the functional relationship between feature variables and model outputs, we clarify the marginal effects of urban morphological indicators on CUHII, which supports the identification of key driving factors and their threshold characteristics.

### 2.3.4 ENVI-met Model setup and initialization

ENVI-met has been widely applied in cooling effect assessment (Di Giuseppe et al., 2021), temperature field prediction (Forouzandeh, 2021), and thermal comfort research (Berardi et al., 2020). The selection of ENVI-met simulation areas in this study was based on two core principles: ① Urban morphological representativeness: Typical functional zones in Beijing were selected, covering dominant urban forms such as high-density high-rises and low-density low-rises, which can reflect the representative spatial characteristics of Beijing's urban area; ② Data support: These zones are equipped with long-term AWS operated by the China Meteorological Administration, which provide continuous air temperature data at a height of 1.5 meters, serving as a reliable benchmark for model validation.

The model integrates high-resolution Google Earth imagery and field survey data to accurately construct the three-dimensional spatial configuration of buildings, vegetation, and soil, with vegetation parameters derived from ENVI-met's 3D plant database. The horizontal extent of the model was set to 1×1 km (200×200 grids, 5 m resolution), with 65 grid layers in the vertical direction. The setting of thermal property parameters for surface materials integrated field sampling analysis and calibration results from existing literature (Meng et al., 2024): ① Impervious surfaces: Dominated by asphalt, with parameters set with reference to the heat conduction and radiation characteristics of typical urban asphalt pavements; ② Pervious surfaces: Mainly composed of loam, with parameters determined based on the heat capacity and thermal conductivity of soil samples from the study area; ③ Vegetation parameters: Set in combination with the leaf radiation characteristics and transpiration parameters of common tree species in Beijing, which affect the surrounding thermal environment through transpiration and shading. To reduce boundary effects, a 10-layer nested grid technique was used (Kong et al., 2016), with surface materials set as a mixture of loam and asphalt. The model's boundary meteorological parameters (temperature, humidity, wind speed, wind direction) were updated every 30 minutes using a complete forcing method, with data obtained from meteorological station measurements. For model validation, the $R^2$ and RMSE were adopted, with a focus on the simulation accuracy of air temperature at a height of 1.5 meters. Typical urban meteorological stations in Beijing were selected, multi-scenario simulation schemes were designed, and emphasis was placed on analyzing the mechanisms by which morphological indicators act on CUHII, canopy ventilation, and radiation exchange.

### 3 Results

### 3.1 Diurnal variations of CUHII during HW periods

Under climate warming, urban expansion has increased built areas, with human activities generating additional anthropogenic heat and pollutant emissions that intensify urban warming. Using observational data from Beijing's AWS, this study examines the diurnal variations of CUHII during both heatwave and non-heatwave periods.

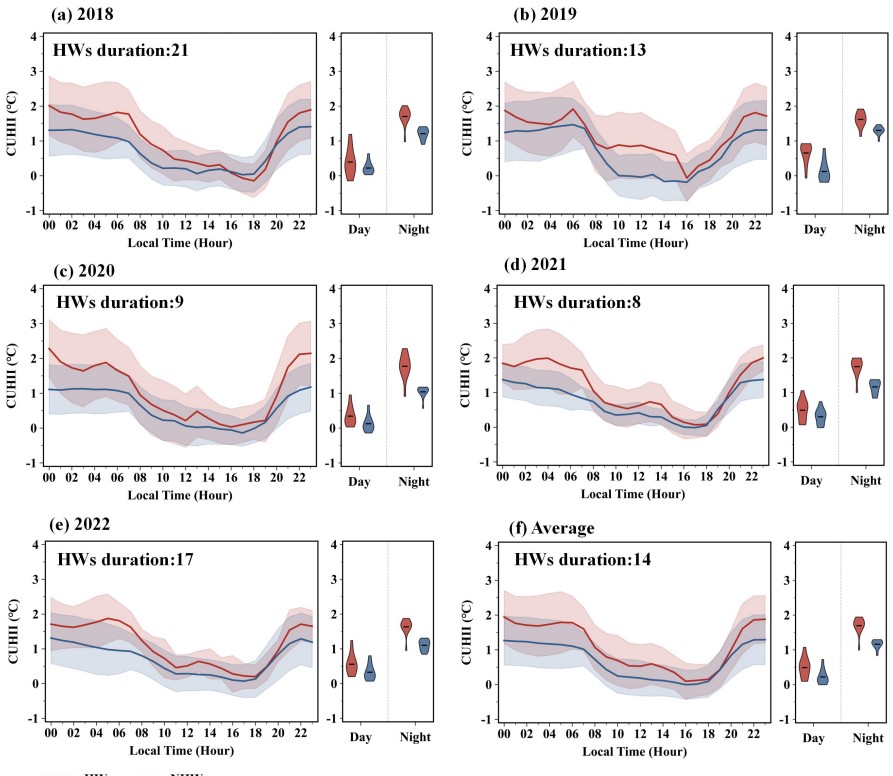

**Figure 2: Diurnal variations of the CUHII during the NHW and HW periods. (a)-(e) Year-specific patterns; (f) Multi-year average. Left panels: CUHII diurnal cycles (solid lines) with shaded areas showing standard deviations. Right panels: Violin plots of CUHII distributions during the day (08:00-19:00) and at night (00:00-07:00, 20:00-24:00).**

In Figure 2, the summer diurnal variations of CUHII in Beijing megacity during HW periods exhibit a U-shaped fluctuation. CUHII begins to decline gradually at 06:00 Beijing Time (BJT), reaches the lowest value at 16:00 BJT, then gradually rebounds, and remains at a high level from 22:00 BTJ to 05:00 BJT. The diurnal variation trend of CUHII during NHW periods is generally consistent with that during HW periods. In particular, except for 19:00 BJT 2018 (Figure 2c), the hourly CUHII values during HW periods in each year are higher than those during NHW periods. From the annual average (Figure 2f), CUHII during the HW periods ranges from 0.18 to 2.06 °C, significantly higher than 0.03 to 1.32 °C during the NHW periods, indicating a significant intensification of CUHII during the HW periods compared to the NHW periods (Cheval et al., 2024; Shi et al., 2024).). The violin plots clearly show the diurnal distribution characteristics of CUHII during the HW (red) and NHW (blue) periods: during the day, the mean CUHII during the HW periods is 0.54 °C, slightly higher than 0.23 °C during the NHW periods; at night, the median CUHII during the HW periods reaches 1.71 °C, with a more

significant increase than 1.12 °C during the NHW periods. It should be noted that during both NHW and HW periods, nighttime CUHII is generally significantly higher than daytime CUHII. This can be explained by the urban-rural differences in energy budgets: during the daytime, cities are heated by solar radiation, with surface heat transferred to the atmosphere through turbulence and regulated by ventilation conditions; at nighttime, urban buildings and impervious surfaces release stored heat, while suburbs form radiative cooling due to vegetation cover, further widening the urban-rural temperature difference (Zhou et al., 2019; Shen et al., 2024). Furthermore, the diurnal variation characteristics of CUHII are not absolute, as their intensity and timing distribution vary with the geographical environment of cities. For example, the CUHII in Shanghai during HW periods and its difference from that in non-heatwave periods are strongest around noon (Ao et al., 2019; Tan et al., 2010), and this pattern has also been verified in Athens, Greece (Founda et al., 2017). Such differences from Beijing (where nighttime CUHII is stronger) mainly stem from variations in local circulation: coastal cities like Shanghai and Athens are significantly affected by sea-land breeze advective cooling, and the large heat capacity of seawater weakens the nighttime urban-rural temperature difference; in contrast, nighttime CUHII in Beijing, an inland city, is mainly dominated by surface radiation budgets (Ao et al., 2019).

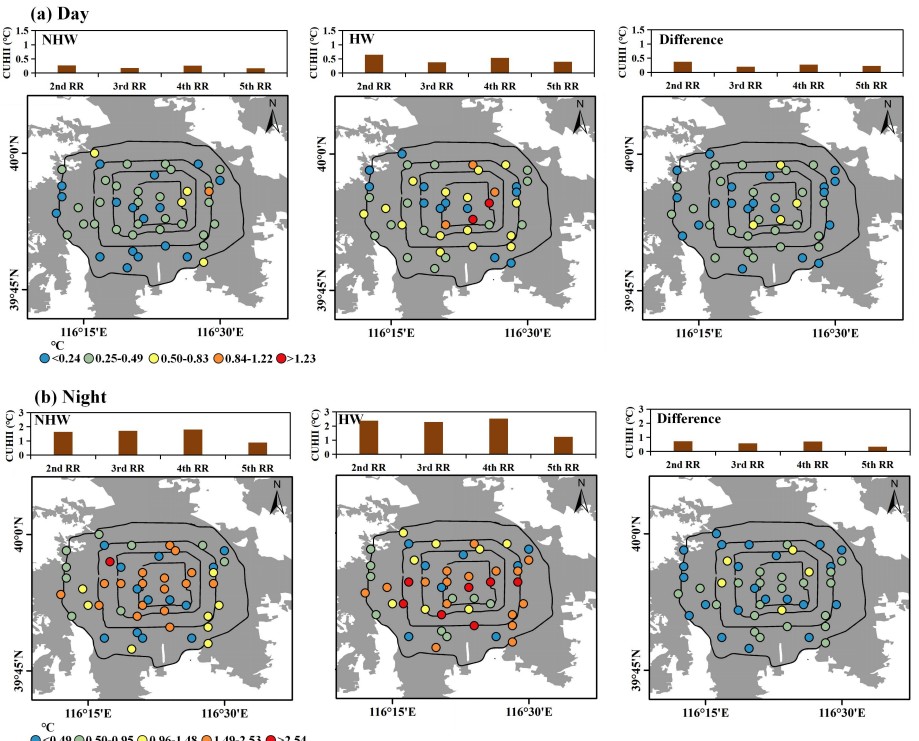

**Figure: 3 Diurnal spatial patterns of CUHII during NHW & HW. Panel (a) for daytime, (b) for nighttime. In each panel, left: NHW CUHII stats & distribution; middle: HW CUHII stats & distribution; right: HW-NHW CUHII difference.**

Beijing megacity has experienced rapid and large-scale urbanization over the past few decades, with urban spaces
continually expanding to the suburbs, leading to a significant CUHI effect (Zheng et al., 2018). Spatial analysis of daytime
CUHII (Figure 3a) reveals that the Second Ring Road exhibits the highest CUHII values across all metrics: 0.27°C during
NHW periods, 0.65°C during HW periods, and a difference of 0.38°C between the two. Analysis of urban configuration
structures (Figure 4a) shows that the Second Ring has the highest proportion of dense buildings, and the compact layout
leads to the accumulation of solar radiation heat in dense building clusters during the day, which is difficult to diffuse (Ge et
al., 2016). This may be an important reason for the increase in daytime CUHII during the HW periods. The nighttime CUHII
differs (Figure 3b), with the Fourth Ring having the highest CUHII (1.80°C during NHW periods, 2.52°C during HW periods,
and a difference of 0.72°C between the two). The Fourth Ring exhibits the highest proportion of high-rise buildings (Figure
4b). The concentrated emission of anthropogenic heat sources, such as air conditioners, in these high-rise zones (Yin & Zhao,
2024) could potentially contribute to the intensification of nighttime CUHII during heatwave events. Thus, urban
morphology may be an important factor for the formation of diurnal patterns of CUHII. In the following sections, this study
will conduct more reliable analyses using machine learning and numerical simulation methods.

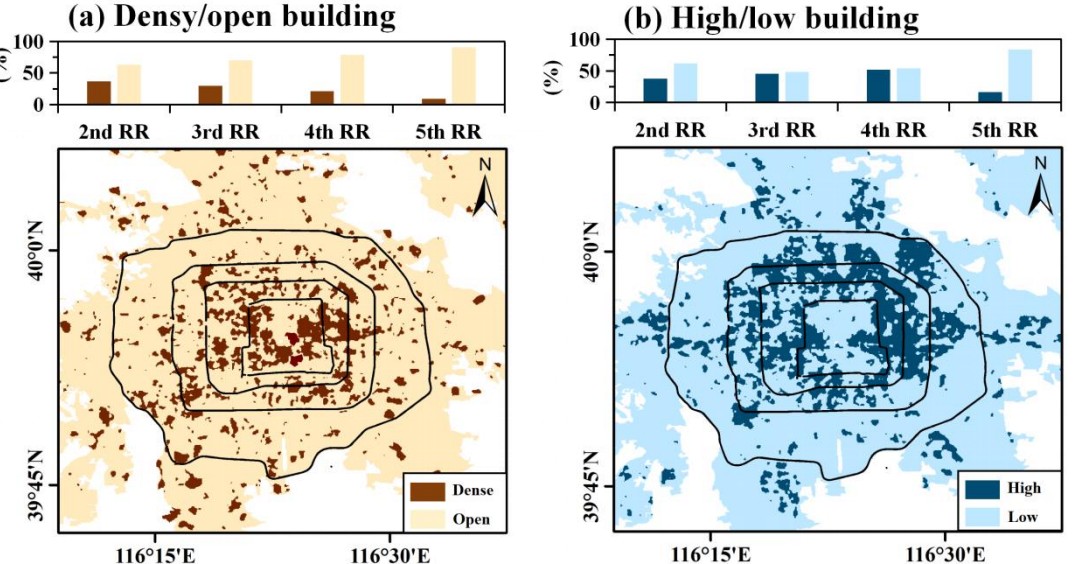

**Figure: 4 (a) Urban configuration structures are dominated by density information, including dense and open rise. (b) Urban
configuration structures are dominated by height information, including high-rise and low-rise.**

### 3.2 Non-linear responses of CUHII to urban morphology

The spatial heterogeneity of urban morphology leads to an uneven distribution of near-surface air temperature by altering the
surface energy balance and heat exchange processes. This section focuses on exploring the influence of urban morphological
indicators on the diurnal spatial patterns of CUHII in Beijing megacity during HW periods.

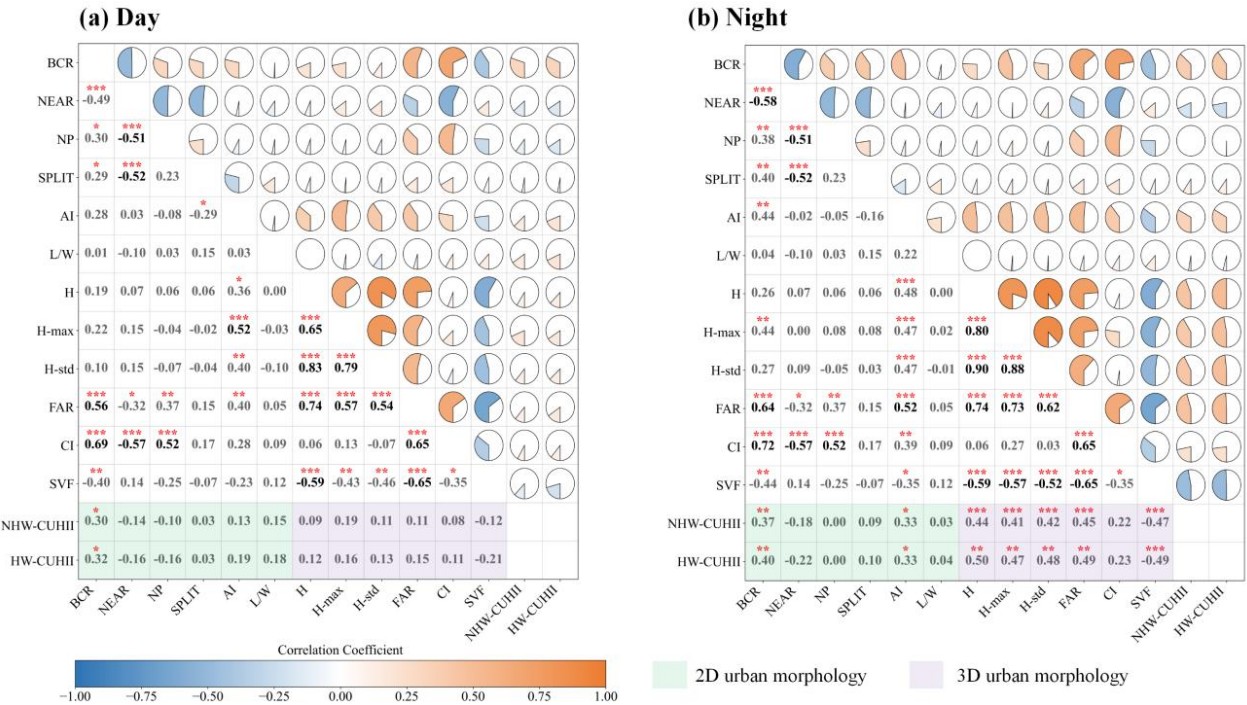

**Figure: 5 Pearson correlation coefficients between urban morphology indicators and diurnal CUHII.**

Before conducting machine learning modeling, we first conducted a preliminary analysis of the linear relationship between urban morphological indicators and CUHII. Figure 5a shows that, during the day, regardless of NHW or HW periods, the BCR among 2D morphological indicators exhibits the strongest correlation with CUHII, and the SVF shows the most significant negative correlation with CUHII among 3D indicators. At night, among the 2D indicators, BCR still shows the highest correlation, while among the 3D indicators, H exhibits the strongest correlation. It is noteworthy that the correlation between 2D indicators and CUHII significantly intensifies: for example, the correlation coefficient between BCR and CUHII increases from 0.37 during NHW periods to 0.40 during HW periods. The influence of 3D morphological indicators is also significantly enhanced (Fig. 5b). The H, H_max, H_std, and FAR all show significant positive correlations with CUHII (r>0.41 during NHW periods, r>0.47 during HW periods). The correlation coefficient between SVF and CUHII increases to -0.47 (NHW periods) and -0.49 (HW periods). These results indicate that daytime CUHII is primarily regulated by the horizontal heterogeneity of urban morphology, while nighttime CUHII is driven mostly by vertical urban morphology. Furthermore, the correlations between morphological indicators and CUHII during the HW periods are generally higher than during the NHW periods.

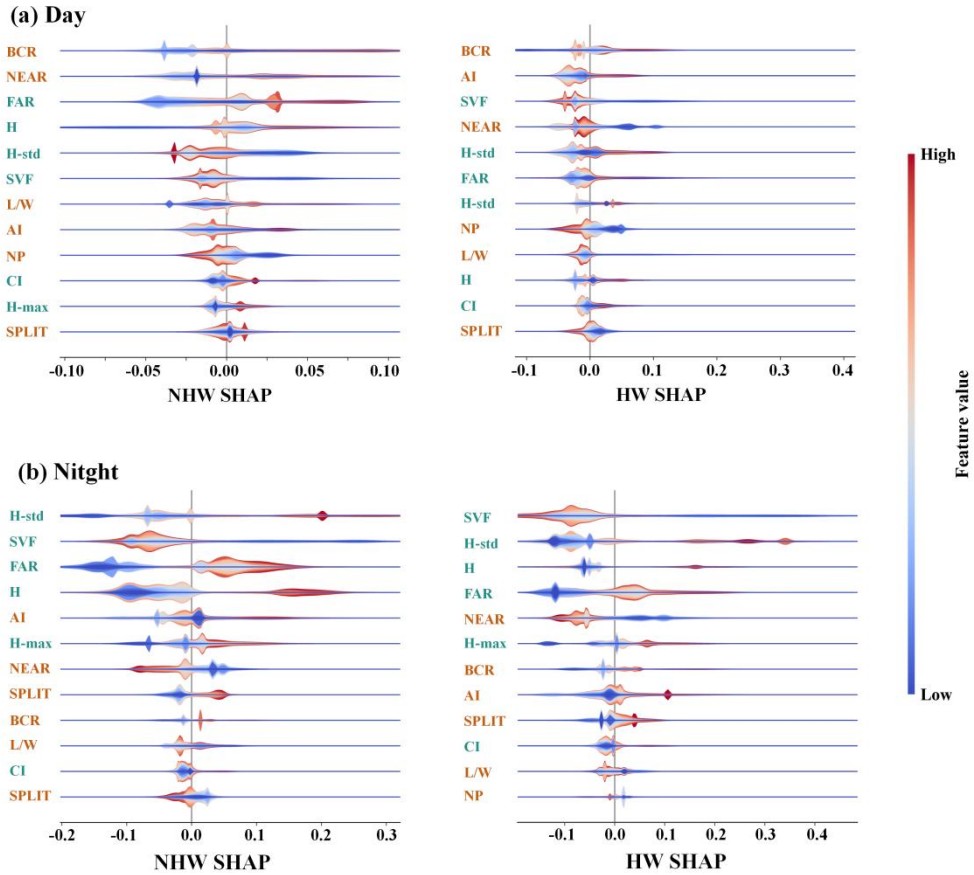

**Figure 6: SHAP value analysis of urban morphology indicators for diurnal CUHII during NHW and HW periods, using XGBoost model. SHAP quantifies feature contributions to model outputs. The red/blue color gradients represent high/low feature values, with red indicating 2D urban morphological indicators and green indicating 3D urban morphological indicators.**

Fig. S1 illustrates the predictive performance of the XGBoost model for CUHII. For the test dataset, all R² values exceed 0.40. Except for nighttime CUHII during HW periods (where the relatively large RMSE is directly linked to the largest intrinsic magnitude of CUHII in this period), the RMSEs in other scenarios are within 0.50 °C. These results indicate that the XGBoost model can be regarded as a reliable tool for fitting the relationship between local thermal environment and urban

morphology (He et al., 2024; Lin et al., 2024). The analyses based on Figures 6 and 7 indicate that: During daytime NHW periods, SHAP values of the 2D indicator BCR concentrate in the positive interval (left panel of Figure 6a), ranking first in importance (left panel of Figure 7a). This is because increased in building density leads to compact layouts and weakens the ventilation potential (Ng et al., 2011; Xu et al., 2019), with NEAR and FAR ranking second and third, respectively, and FAR showing a wider range of values. During daytime HW periods, the positive concentration trend of BCR becomes more

significant (right panel of Figure 6a), maintaining its importance top- (right panel of Figure 7a). AI ranks second, while SVF shows an obvious negative deviation, consistent with the weak negative correlation observed in Figure 4 during the day.

Furthermore, the mean SHAP values of the 2D indicators during the day are higher than those of the 3D indicators. Compared to NHW periods, the mean importance of 2D and 3D indicators during HW periods increases by 35.4% and 36.7%, respectively. At night during NHW periods, the dominance of 3D indicators begins to emerge: the SHAP value range

of H_std expands significantly (left panel of Figure 6b), rising to the top in importance (left panel of Figure 7b), followed by SVF and FAR in the second and third positions, respectively. The BCR drops to the 9th position but remains positive. During nighttime HW periods, the dominance of 3D indicators is further enhanced: the SHAP value range of SVF expands (right panel of Figure 6b), stably ranking first in importance (right panel of Figure 7b). High-rise residences, accompanied by high population density and air conditioning heat dissipation, exacerbate the heat island effect (Ryu & Baik, 2012).

Compared to NHW periods, the mean importance of 2D and 3D indicators during HW periods increases by 16.2% and 31.3%, respectively.

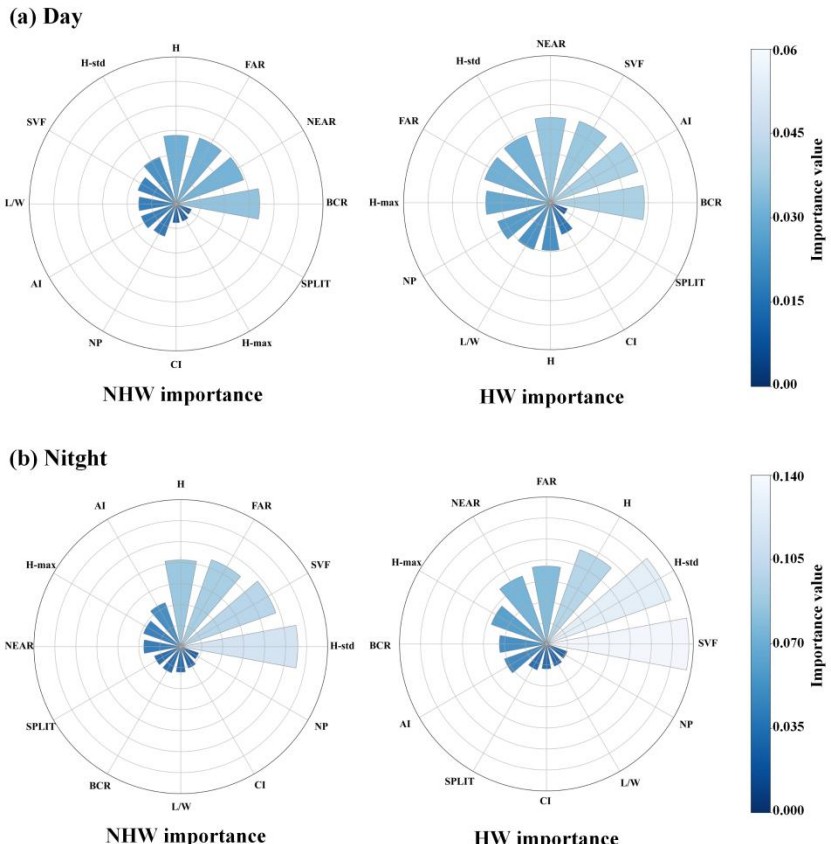

**Figure 7: Importance value analysis of urban morphology indicators for diurnal CUHII during NHW and HW periods, using XGBoost Model. This radar - plot visualization leverages XGBoost's feature importance algorithm, quantifying how urban**

**morphology indicators influence CUHII. It distinguishes day/night responses and NHW/HW scenarios, with color gradients (blue - to - light) representing increasing importance values.**

Figure 8 reveals the dependency characteristics of urban morphology on CUHII: During the day, BCR shows nonlinear positive driving (left subplot of Figure 8a), with a significant threshold effect in the low-coverage interval (<0.12). The positive contribution growth rate slows when the BCR exceeds this value, and the threshold effect is more prominent during HW periods (Guo et al., 2016; Yang et al., 2018). SVF shows a negative effect in the interval of 0.725-0.735 and turns positive in the interval of 0.735-0.75 (middle subplot of Figure 8a), which may be related to the dual role of the height of the building in the thermal environment (Perini & Magliocco, 2014). Two-factor analysis shows (right subplot of Figure 8a) that CUHII reaches its peak (yellow area) when BCR $\geq$ 0.23 and SVF $\leq$ 0.72, indicating that high BCR and low SVF synergistically exacerbate CUHII.At night, the increase in CUHII with the rise of BCR is more gentle than during the day (left subplot of Figure 8b), without obvious abrupt nodes, and the dominance of BCR weakens. SVF only shows negative regulation, and its intensity is higher than during the day (middle subplot of Figure 8b). Two-factor analysis indicates (right subplot of Figure 8b) that CUHII is the highest (yellow area) when BCR $\geq$ 0.23 and SVF $\leq$ 0.72, and the area of this region expands during HW periods. When SVF $\geq$ 0.75, the increase in BCR has a limited impact on CUHII, suggesting that high SVF can mitigate CUHII. In summary, the regulation of urban morphology in CUHII exhibits significant diurnal asymmetry: 2D indicators dominate during the daytime, while 3D indicators dominate at night. HW events can improve the non-linear modulation of urban morphological indicators. In summary, the regulation of urban morphology on CUHII exhibits significant diurnal asymmetry: 2D indicators predominate during the daytime, while 3D indicators play a dominant role at night. Furthermore, urban morphology exerts nonlinear modulation on CUHII, characterized by threshold effects and dual roles (e.g., SVF showing both negative and positive impacts), with these nonlinear effects being more pronounced during HW periods.

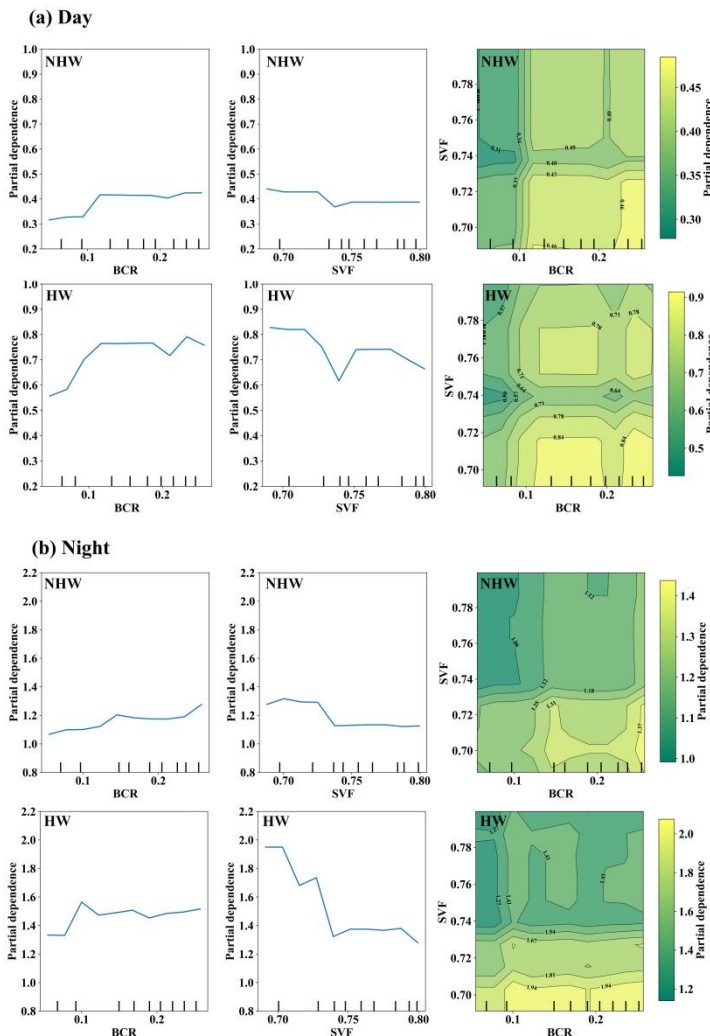

**Figure 8: Daytime and nighttime CUHII PDP on urban morphology: the left subplot shows the PDP for BCR, the middle subplot shows the PDP for SVF, and the right subplot shows the two-way partial dependency plots for BCR and SVF.**

### 3.3 Simulation of microclimate effects of key urban morphological indicators

This section draws on previous microclimate studies based on building morphology (Hu et al., 2022; Nugroho et al., 2022) and ensures scenario stability, using SVF as the 3D morphological indicator to conduct multi-scenario simulations via ENVI-met. The analysis focuses on exploring the influence mechanisms of SVF on the diurnal variations of local urban local

environments.

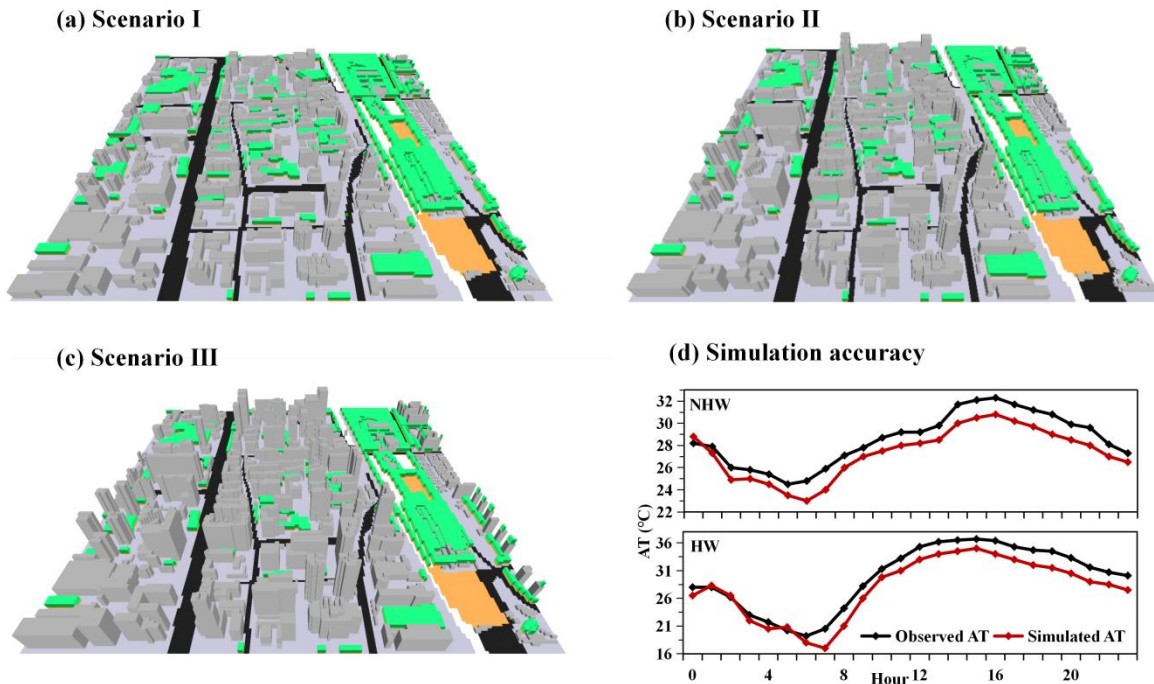

**Figure 9: Urban morphology scenarios and simulation validation in the Study Area. (a) baseline real world scenario, the color scheme in distinguishes vegetation, buildings, and open spaces; (b-c) modified scenarios with reduced SVF; (d) simulation accuracy of air temperature (AT) for Scenario I during NHW and HW periods.**

This section selected a 500-meter radius area around Station 651061 on the North Fourth Ring Road as the simulation region, where the BCR was 0.225 and the SVF was 0.76. Three scenarios were set up by adjusting building heights (with street width, building area kept unchanged to isolate the independent effect of SVF): ① Scenario I: Used the original building heights in the study area, corresponding to the real SVF (0.76, Figure 9a); ② Scenario II: Based on the PDP analysis results

of the machine learning model, building heights were adjusted to reduce SVF to 0.735 (the critical point of positive/negative effects, Figure 9b); ③ Scenario III: Building heights were further adjusted to reduce SVF to 0.685 (the rapid growth stage of negative effects, Figure 9c). Notably, building height modifications were applied uniformly across the entire simulation domain to ensure consistent spatial conditions except for SVF differences. As indicated in Figure 9d, based on Scenario I, the ENVI-met model effectively simulated the diurnal variations of air temperature on days of NHW days (17 June  2020)

and days of HW (15 June 15): the $R^2$ and RMSE for observed versus simulated air temperature (AT) were 0.64 and 1.25℃ on NHW days, and 0.73 and 1.16℃ on HW days, respectively. Compared with the findings of Morakinyo et al. (2018) and Tan et al. (2017), the correlation between simulated and observed values and the trend of temporal changes showed consistency. Due to the simplified treatment of building material heat capacity and environmental thermal radiation processes in the model (Ali-Toudert & Mayer, 2006), the simulated air temperatures were generally lower than the measured

values during the daytime. The difference between simulation and observation gradually narrowed after sunset, and the

simulation error exceeded the measured value during 02:00–04:00. Overall, the trends of simulated and measured air temperature variation showed high consistency, indicating that the model could effectively reflect the diurnal characteristics of local urban thermal environments. Due to the simplified treatment of the heat capacity of building materials and environmental thermal radiation processes in the model (Ali-Toudert & Mayer, 2006), simulated air temperatures were generally lower than measured values during the daytime. The difference between simulated and observed air temperatures gradually narrowed after sunset, and simulation errors exceeded measured values between 02:00–04:00. Compared with the findings of Morakinyo et al. (2018) and Tan et al. (2017), the simulated AT in this study could effectively reflect the diurnal variations of the urban local thermal environment.

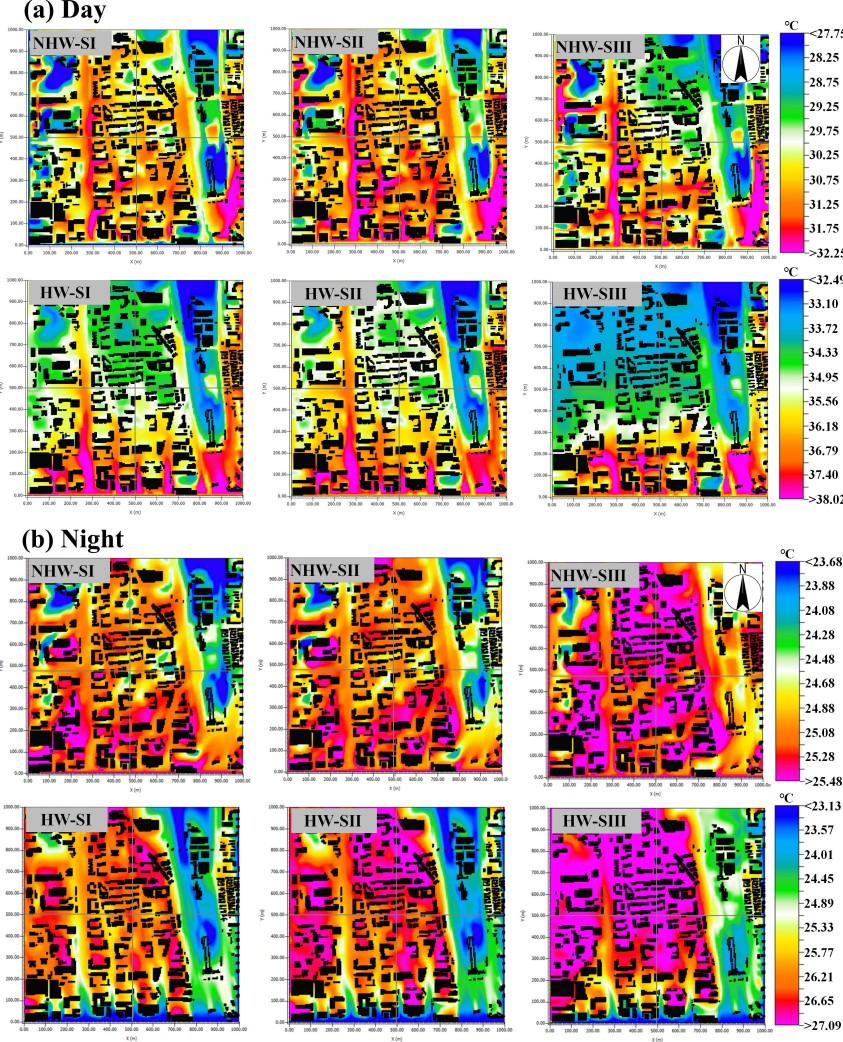

**Figure 10: Spatial distributions of simulated AT across scenarios during daytime (a) and nighttime (b). NHW-SI represents Scenario I during NHW periods, HW-SI represents Scenario I during HW periods, and so forth. The intersection of the two gray crosshairs in each subplot indicates the location of the meteorological station.**

The figure above shows the simulated AT spatial distribution under different scenarios during daytime (Figure 10a). Spatial patterns reveal that during NHW periods, Scenario II shows a 0.2–0.7°C temperature rise across the study region. The central point confirms this trend, with AT increasing from 30.68°C in Scenario I to 31.09°C in Scenario II. Meanwhile, Scenario III exhibits a 0.3–0.8°C cooling in these areas, driven by building shadows, with the central point AT in Scenario III decreasing to 30.33°C. During HW periods, these effects intensify. Scenario II sees a 0.5–1.1°C warming across these zones, with the central point air temperature in Scenario II increasing from 35.01°C to 35.76°C. Scenario III shows a 0.6–1.4°C cooling in study region, with the central point AT in Scenario III dropping to 34.39°C. As SVF decreased, the obstruction of building clusters to air flow intensified, reducing the heat dissipation capacity. Meanwhile, blocking of long wave radiation was exacerbated, promoting heat accumulation and leading to temperature increases. It should be noted that the temperature change patterns in Scenario III, like the drop in central point AT, are related to excessively low SVF significantly increasing building shadow areas, enhancing the shading effect on solar radiation, thus reducing surface heat absorption and inhibiting temperature rise (Perini & Magliocco, 2014). Figure 10b shows the spatial distribution of the simulated AT indifferent scenarios at night. During NHW periods, the central point AT in Scenario I was 24.86°C, increasing to 25.10°C in Scenario II with a relatively small variation, while that in Scenario III increased significantly to 25.90°C. During HW periods, the central point AT in Scenario I was 26.25°C, increasing to 26.83°C in Scenario II and increased significantly to 27.93°C in Scenario III. Notably, this pattern of temperature variation (moderate rise in Scenario II, sharp increase in Scenario III) is consistent across the entire simulation domain. The increase in building height hinders the convective heat dissipation of nighttime air, making heat dissipation difficult and thus promoting a significant temperature rise (Mo et al., 2024). Furthermore, the temperature differences between the scenarios during the HW periods were more significant than during the NHW periods, indicating that changes in building height have a more pronounced impact on air temperature during the HW periods, further amplifying the non-linear modulation of the building SVF in AT.

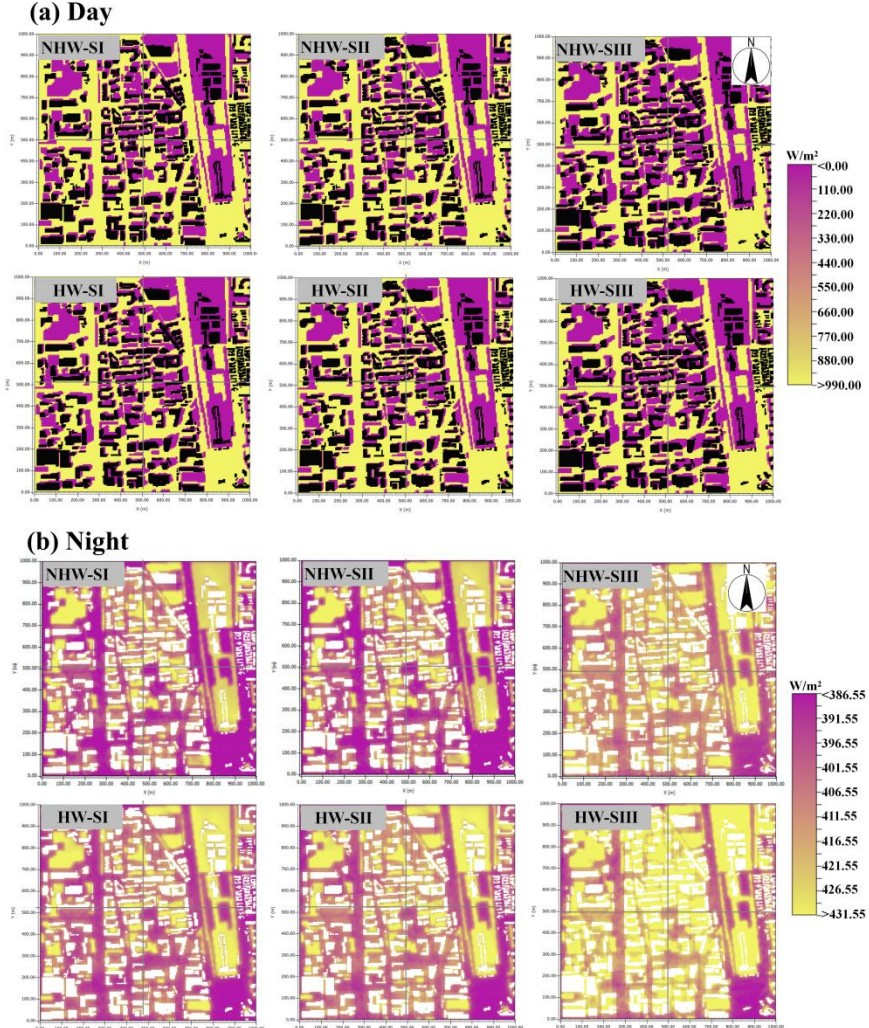

**Figure: 11 Spatial distribution of simulated short-wave (SW) radiation (a) and long-wave (LW) radiation (b) across scenarios during NHW and HW periods.**

Combined with the spatial distribution of short-wave (SW) radiation, the temperature phenomena under different SVF daytime conditions can be further explained (Figure 11a). Overall, SW radiation during HW periods is higher than during NHW periods. Specifically, in Scenario II during the HW periods, the average SW radiation slightly decreases from 636.16 W/m² to 602.27 W/m², the SW radiation at the central point decreases from 970 W/m² to 930 W/m², but AT shows an upward trend. This can be attributed to the obstruction of air flow by buildings (Ge et al., 2025), where the heat

accumulation effect dominates in the competition between SW radiation attenuation caused by increased building height and air flow resistance. In Scenario III, the average SW radiation drops to 537.88 W/m², the central point's SW radiation plummets to 860 W/m², and significant shadow shading leads to a substantial reduction in SW radiation (Lin et al., 2024),

thereby inhibiting the temperature rise. At night, the heat dissipation of LW radiation exhibits stronger non-linear threshold characteristics (Figure 11b). In Scenario II during the HW periods, the average LW radiation increases from 408.34 W/m² to 412.81 W/m², and the LW radiation at the central point climbs from 388 W/m² to 394 W/m². At this time, the resistance to escape of LW radiation is limited, so the air temperature only increases slightly. In Scenario III, the lower SVF significantly reduces the loss of LW radiation to the atmosphere, with the average LW radiation rapidly increasing to 424.31 W/m², and the central point's LW radiation surges to 410 W/m², accompanied by a noticeable temperature increase. This is because multiple reflections between building facades retain radiation energy within urban canyons, thus enhancing the capture of LW radiation (Mei et al., 2025). In summary, buildings exert nonlinear modulation on urban diurnal thermal environments through the competitive effects of SW radiation shading and ventilation resistance, as well as the reflection and accumulation mechanisms of LW radiation.

## 4 Discussions

Urban morphology influences the CUHI by altering surface properties and spatial structures. As a dynamic meteorological factor, the inherent relationship between the wind field and CUHII should not be overlooked. This section analyzes the modulation mechanisms of the wind field on the diurnal CUHII during HW periods.

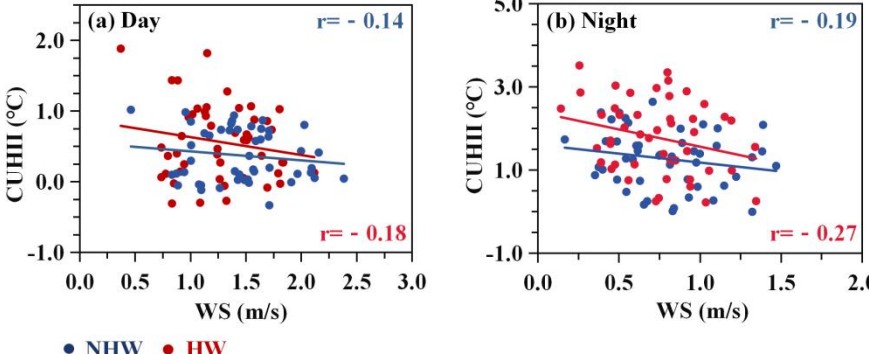

**Figure 12: Correlation between wind speed (WS) and CUHII during NHW (blue dots) and HW (red dots) periods. Subplots show daytime (a) and nighttime (b) relationships.**

Figure 12a shows that during the daytime, the correlation coefficients (r) between WS and CUHII were -0.14 during NHW periods and -0.18 during HW periods, indicating a weak negative correlation that was slightly stronger during HW periods. Deng et al. (2025) simulated that a 10% increase in WS could reduce the CUHII by 0.16°C during summer days. Stronger solar radiation during HW periods makes the heat dissipation effect of wind more significant for CUHII. During the night (Figure 12b), the r was -0.19 during NHW periods and -0.27 during HW periods, with enhanced negative correlations compared to daytime, especially during HW periods. This may be related to the heat dissipation characteristics of the underlying urban surface during nighttime (Liu et al., 2022), where slower heat release makes the modulation of WS in

CUHII more pronounced. Notably, compared with research findings from other cities (Yang et al., 2023; Rajagopal et al., 2023; Deng et al., 2025), the CUHII in Beijing exhibits a unique characteristic—it is insensitive to WS variations both
during the daytime and nighttime. This phenomenon may be closely linked to urban morphology and local geographical environments. Urban morphology significantly modulates wind penetration and heat exchange efficiency: compact built-up areas with high BCR and low SVF (e.g., the Second Ring Road) form dense building clusters that block airflow, reducing WS and weakening wind-driven heat dissipation, thus making CUHII less responsive to WS changes. In addition, existing studies have confirmed that local circulations formed under different geographical backgrounds can significantly reshape the
spatiotemporal distribution of urban extreme high temperatures (Zhang et al., 2011; Zhou et al., 2020; Chen et al., 2022). Specifically for Beijing, the mountainous terrain in its western and northern regions gives rise to a typical mountain-valley wind circulation, which interacts with urban morphology: dense buildings in central areas disrupt valley breeze penetration, while sparse layouts in suburbs align with mountain winds. This interplay between morphology and terrain-induced winds weakens the modulation of WS variations on CUHII. Observations show that wind directions in Beijing's urban area display
a regular diurnal variation: northerly winds (mountain breeze) dominate from 05:00 to 10:00 Beijing Time; there is an obvious reversal around 11:00, shifting to southerly winds (valley breeze) which persist until 04:00 the next day. Additionally, the average WS of the mountain breeze is significantly lower than that of the valley breeze (Zheng et al., 2018). Such distinct periodic characteristics make mountain-valley breeze a key local factor influencing Beijing's thermal environment (Dou et al., 2014). Based on this, we speculate that the "insensitivity of CUHII to WS variations" observed in
this study may be the result of interactions between the mountain-valley breeze cycle, urban morphology, and the inherent diurnal cycle of CUHII.

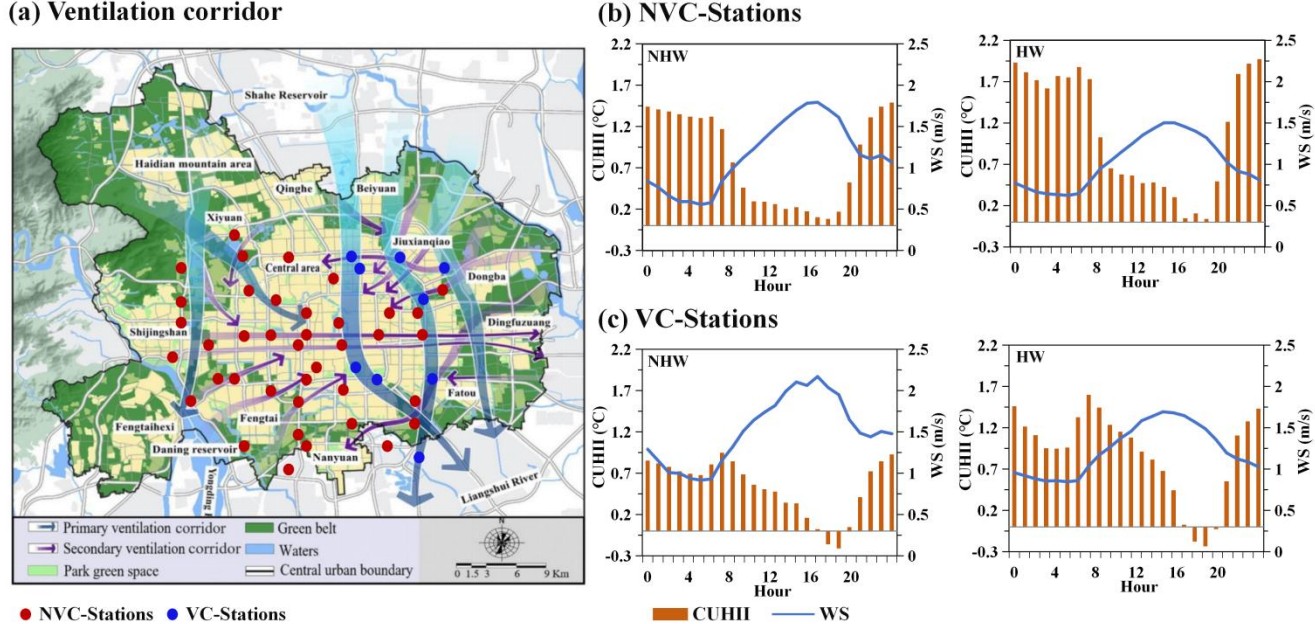

**Figure 13: Impacts of ventilation corridors on diurnal variations of WS and CUHII. (a) Urban ventilation corridor planning in Beijing. Based on the Beijing Urban Master Plan, published by the Beijing Municipality Government. (b) Diurnal variations of WS and CUHII during NHW and HW periods at Non-Ventilation Corridor Stations (NVC-Stations). (c) Diurnal variations of WS and CUHII during NHW and HW periods at Ventilation Corridor Stations (VC-Stations).**

Urban ventilation corridors represent an energy-efficient ecological approach to improving urban wind-thermal environments by taking advantage of natural meteorological conditions (Masmoudi & Mazouz, 2004; Masson, 2006; Palusci et al., 2021). In recent years, Beijing has proposed to construct ventilation corridors to alleviate increasingly severe urban environmental problems, with corridor designs intentionally aligned with urban morphological features—such as low BCR, high SVF, and wide street canyons—to minimize aerodynamic resistance (Figure 13a). This section designates nine stations within first-level ventilation corridors (VC-Stations) as those embedded in open built-up areas (sparse buildings, low-rise structures) and the remaining 39 stations in compact built-up areas (NVC-Stations) as Non-Ventilation Corridor Stations. Data show that WS at NVC-Stations (Figure 13b) is significantly lower than that at VC-Stations (Figure 13c), a difference primarily driven by urban morphological controls: dense high-rises in NVC areas disrupt airflow, while VC areas' open layouts allow unobstructed wind penetration. For example, at night during HW periods, WS at NVC-Stations remains around 0.5 m/s due to wind blockage by closely packed buildings, whereas that at VC-Stations stays above 0.8 m/s, facilitated by their low-rise, sparse morphologies. CUHII in VC-Stations generally exhibits an inverse relationship with WS, with morphological traits amplifying this effect. At NVC-Stations, their compact morphologies (high BCR, low SVF) limit heat dissipation; when WS is 0.5 m/s in the early morning during HW periods, CUHII reaches 1.9°C due to trapped heat. In contrast, when WS increases to 1.5 m/s in the afternoon at VC-Stations—where open morphologies enhance turbulent heat exchange—CUHII drops to only 0.3°C. Notably, the CUHII mitigation effect of ventilation corridors shows significant diurnal differences influenced by urban morphology. During the daytime, high baseline WS reduces the relative impact of ventilation corridor-induced WS gains, but VC areas' low-rise structures still promote more efficient heat dispersion than NVC's dense canyons. During nighttime, with lower background WS, the WS enhancement from VC's open morphologies is more pronounced (Hsieh & Huang, 2016), and the thermal environment—sensitive to trapped heat in NVC's compact morphologies—is more responsive to WS modulation (She et al., 2022), resulting in significantly lower nighttime CUHII at VC-Stations (42.09% lower during NHW periods and 33.91% lower during HW periods).

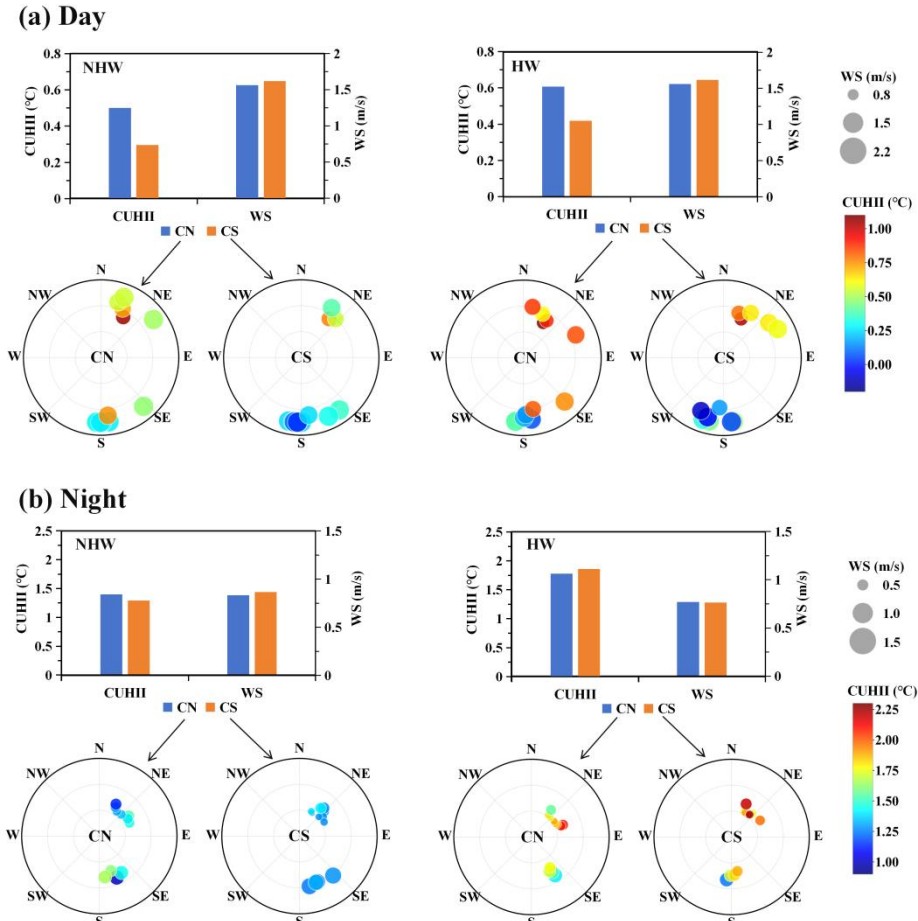

**Figure 14: Diurnal CUHII and wind rose diagrams for city northern (CN) stations and city southern (CS) stations: (a) daytime; (b) nighttime. In the wind rose diagrams, bubble positions indicate wind direction, bubble sizes represent WS magnitudes, and bubble color intensity reflects CUHII strength.**

The bar charts in Figure 14a show that during the daytime, the differences in WS between the CN (city northern) and CS (city southern) stations are minimal during both the NHW and HW periods. However, CUHII at CN stations is 0.49°C during NHW and 0.61°C during HW, significantly higher than 0.28°C and 0.42°C at CS stations, indicating that WS might not be the primary cause spatial patterns of CUHII. Wind transports heat through thermal advection in urban areas (Wang et al., 2020), potentially exacerbating the risks of thermal exposure in specific regions (Heaviside et al., 2015; Bassett et al., 2016). During the daytime, the predominant southerly winds (approximately 0.75 m/s) promotes horizontal heat transport from the upstream to downstream urban areas, increasing CUHII in northern urban regions. At night (Figure 14b), predominant northerly winds (approximately 0.75 m/s) prevail, and the CUHII shows little difference between CN and CS stations. This phenomenon can be attributed to the WS threshold effect of horizontal heat transport. Weak prevailing winds

stabilize the atmospheric stratification, thereby hindering urban heat dissipation, particularly during clear and light-wind nights (Kim & Baik, 2005). Studies further indicate that the CUHI center drifts downwind with increasing WS, with an average drift speed threshold—when WS is below this threshold, spatial differences in the urban thermal environment are primarily determined by local underlying surface properties (e.g., green space ratio, building density) (Xu et al., 2019). In summary, the impact of wind field on CUHII is jointly influenced by WS, direction, and source, with diurnal differences in modulation mechanisms: nighttime wind significantly mitigates CUHII, especially in ventilation corridor areas, while daytime prevailing winds may exacerbate thermal burdens in downstream regions through thermal advection rather than serving as simple cooling factors.

## 5 Conclusions

By integrating ground observations, XGBoost, and ENVI-met simulations, this study systematically unravels the diurnal asymmetric and nonlinear response of canopy urban heat island (CUHI) to urban morphology during heat wave (HW) periods in Beijing. The results show that compared with non-heat wave (NHW) periods, CUHI intensity (CUHII) during HW periods is significantly enhanced, with a 91.3% increase in daytime and 52.7% at night, and its diurnal variation presents a U-shaped fluctuation with distinct spatial patterns (strongest within the Second Ring Road in daytime and most prominent around the Fourth Ring Road at night). Machine learning analysis indicates that building coverage ratio (BCR) is the most critical driver of daytime CUHII, while sky view factor (SVF) dominates at night; the mean importance of 2D/3D morphological indicators increases by 16.2%–36.7% during HW periods, with significant interactions between BCR and SVF. ENVI-met simulations further confirm the nonlinear modulation mechanism of urban morphology: when SVF decreases from 0.735 to 0.685, daytime temperature regulation is jointly affected by short-wave radiation shading and ventilation resistance, showing a "first warming then cooling" pattern, while nighttime temperature changes are dominated by the reflection and accumulation of long-wave radiation by buildings, exhibiting accelerated warming characteristics. Additionally, the study identifies diurnal differences in the impact of wind fields on CUHII: ventilation corridors can reduce nighttime CUHII by 33.91%–42.09% to mitigate heat islands effectively, whereas daytime prevailing winds may intensify CUHII in downstream regions through thermal advection rather than simply acting as a cooling factor. These findings clarify the diurnal asymmetric formation mechanism of CUHI during HW periods and provide quantitative references for optimizing urban morphology and planning ventilation corridors, offering precise scientific guidance for mitigating urban thermal risks.

**Data availability.** The hourly AWS observation data are available upon request from the China Meteorological Data Service Center (http://data.cma.cn/en). The land cover data are available at https://zenodo.org/record/5816591 (Yang & Huang, 2021).

**Author contributions.** TS and YY conceptualized the study. TS wrote the original manuscript and plotted all the figures.
YY, PQ and SL assisted in the conceptualization and model development. All the authors contributed to the manuscript preparation, discussion, and writing.

**Financial support.** This research has been supported by the National Natural Science Foundation of China (grant nos. 42222503, 42030606, and 42105147), the Anhui Provincial Natural Science Foundation (2508085MD088), the Fund of the Key Laboratory of Transportation Meteorology, China Meteorological Administration & Nanjing Joint Institute for
Atmospheric Sciences (grant no. BJG202506), and the Anhui Key Laboratory of Real Scene Geographical Environment Open Fund Project (grant no. 2024PEG010).

**Competing interests.** The contact author has declared that none of the authors has any competing interests.

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
