# Peer review of "Diurnal Asymmetry in Nonlinear Responses of Canopy Urban Heat Island to Urban Morphology in Beijing during Heat Wave Periods"

_EGUsphere, 2025_

## Author Comment (AC1)

**Response to Review Comments of the First Reviewer**

Dear Reviewer and Editors:

We are sincerely grateful to the editor and reviewer for their valuable time for reviewing our manuscript. The comments are very helpful and valuable, and we have addressed the issues raised by the reviewer in the revised manuscript. Please find our point-by-point response (in blue text) to the comments (in black text) raised by the reviewer. We have revised the paper according to your comments (highlighted in red text of the revised manuscript).

Sincerely yours,

Dr. Yuanjian Yang, representing all co-authors

**Major comments:**

**1. Section 2.2.2: Six 2D and 3D indicators are selected as predictor variables for CUHI, but there can be more indicators. Could authors justify why these indicators are used? A review on morphology variables used in previous regression/ML methods is needed.**

***Response:*** We apologize for the vague description. As requested, we have supplemented a review of morphology variables used in previous study. Please refer to Lines 96-116 for the revised content:

[revised manuscript text omitted]

**2. Section 2.3.1: How many HW days are found based on the criteria used in this study? This information can be put in Figure 2 to better illustrate the length of HWs.**

*Response:* Thank you for your excellent suggestion. We have added the annual duration of HW periods to Figure 2 as recommended. In addition, we have attached a table showing the monthly duration of HW periods for each year in this response. Thank you again for your valuable input.

[Figure]

**Figure 2: Diurnal variations of the CUHII during the NHW and HW periods. (a)-(e) Year-specific patterns; (f) Multi-year average. Left panels: CUHII diurnal cycles (solid lines) with shaded areas showing standard deviations. Right panels: Violin plots of CUHII distributions during the day (08:00-19:00) and at night (00:00-07:00, 20:00-24:00).**

**Tab. R1 The duration of HW periods by year.**

| HWs period duration (day) | 2018 | 2019 | 2020 | 2021 | 2022 |
|---|---|---|---|---|---|
| Jun | 10 | 0 | 3 | 5 | 10 |
| Jul | 6 | 13 | 3 | 3 | 4 |
| Aug | 5 | 0 | 3 | 2 | 3 |
| Sum | 21 | 13 | 9 | 8 | 17 |

**3. Section 2.3.3: The training process of XGBoost model requires more details. What data is used as training, validation, and test set, respectively? How is the model performance evaluated? This is the major flaw because the results in Fig.**

**5 and Fig. 6 will be significantly affected by the model performance.**

*Response:* We have supplemented these important details in the text, including specific information on the data used for the training set, validation set, and test set, as well as the methods for evaluating model performance. In addition, we have added a performance graph of the XGBoost model in predicting CUHII in the supplementary. Please refer to Lines 143-150 and 259-261for the revised content:

"In this study, we first performed iterative calculations on 7 commonly used hyperparameters (eta, gamma, max_depth, min_child_weight, subsample, colsample_bytree, and nrounds) within a preset hyperparameter tuning space, and selected the optimal hyperparameter combination that minimizes model error using a 5-fold cross-validation method (Yang et al., 2020; Lin et al., 2024). After completing hyperparameter optimization, we randomly split the sample points in the Yangtze River Basin at a 7:3 ratio to obtain training samples (70%) and validation samples (30%), which were used for training and validating the XGBoost model, respectively. Meanwhile, the coefficient of determination ($R^2$) and root mean square error (RMSE) were chosen as evaluation metrics for simulation accuracy. "

"Fig. S1 illustrates the performance of the XGBoost model in predicting CUHII. For the test dataset, the $R^2$ values all exceed 0.45, while the RMSE values are all within 0.05. These results indicate that the XGBoost model can be regarded as a reliable tool for fitting the relationship between CUHII and urban morphology (He et al., 2024; Lin et al., 2024)."

[Figure]

**Figure S1:** The performance graph of the XGBoost model in predicting CUHII.

**Reference:**

He, J., Shi, Y., Xu, L., Lu, Z., Feng, M., Tang, J., & Guo, X.: Exploring the scale effect of urban thermal environment through XGBoost model, Sustainable Cities and Society, 114, 105763, https://doi.org/10.1016/j.scs.2024.105763, 2024.

Lin, Z., Xu, H., Han, L., et al.: Day and night: Impact of 2D/3D urban features on land surface temperature and their spatiotemporal non-stationary relationships in urban building spaces, Sustainable Cities and Society, 108, 105507, https://doi.org/10.1016/j.scs.2024.105507, 2024.

Yang, L., Xu, H., & Yu, S.: Estimating PM2.5 concentrations in Yangtze River Delta region of China using random forest model and the Top-of-Atmosphere reflectance, J. Environ. Manag., 272, 111061, 2020.

**4. Section 2.3.4: How did authors select study areas for ENVI-met simulations? And what are the values used for various thermal properties in the model setup?**

*Response:* Thank you for your insightful comments. We apologize for the lack of clarity regarding the selection of study areas and thermal property parameters in the ENVI-met model setup. We have supplemented relevant details in Section 2.3.4, and the revised content is as follows:

[revised manuscript text omitted]

**5. Line 177: the larger nighttime CUHI than daytime CUHI shall be better explained. there have been many studies in the literature, and it will be good to have at least some comparisons against CUHI during HW at different cities.**

*Response:* Thank you for your valuable suggestion. We have strengthened the explanation of the causes for the diurnal differences in CUHII in the manuscript and added comparisons with the diurnal variation characteristics of CUHII in cities such as Shanghai and Athens. The revised content can be found in lines 199–211:

"It should be noted that during both NHW and HW periods, nighttime CUHII is generally significantly higher than daytime CUHII. This can be explained by the urban-rural differences in energy budgets: during the daytime, cities are heated by solar radiation, with surface heat transferred to the atmosphere through turbulence and regulated by ventilation conditions; at nighttime, urban buildings and impervious surfaces release stored heat, while suburbs form radiative cooling due to vegetation cover, further widening the urban-rural temperature difference (Zhou et al., 2019; Shen et al., 2024). Furthermore, the diurnal variation characteristics of CUHII are not absolute, as their intensity and timing distribution vary with the geographical environment of cities. For example, the CUHII in Shanghai during HW periods and its difference from that in non-heatwave periods are strongest around noon (Ao et al., 2019; Tan et al., 2010), and this pattern has also been verified in Athens, Greece (Founda et al., 2017). Such differences from Beijing (where nighttime CUHII is stronger) mainly stem from variations in local circulation: coastal cities like Shanghai and Athens are significantly affected by sea-land breeze advective cooling, and the large heat capacity of seawater weakens the nighttime urban-rural temperature difference; in contrast, nighttime CUHII in Beijing, an inland city, is mainly dominated by surface radiation budgets (Ao et al., 2019)."

Reference:

Founda, D., Santamouris, M.: Synergies between urban heat island and heat waves in Athens (Greece), during an extremely hot summer (2012), Scientific Reports, 7(1), 10973, 2017. 10.1038/s41598-017-11407-6

Tan, J., Zheng, Y., Tang, X., Guo, C., Li, L., Song, G., Zhen, X., Yuan, D., Kalkstein, A. J., F Li:
The urban heat island and its impact on heat waves and human health in Shanghai,
International Journal of Biometeorology, 54, 75–84, 2010. 10.1007/S00484-009-0256-X

Ao, X., Wang, L., Zhi, X., Gu, W., Yang, H., Li, D.: Observed synergies between urban heat
islands and heat waves and their controlling factors in Shanghai, China, Journal of Applied
Meteorology and Climatology, https://doi.org/10.1175/jamc-d-19-0073.1, 2019.

Zhou, D., J Xiao,S Bonafoni,C Berger,Deilami, Kaveh,Zhou, Yuyu,Frolking, Steve,Yao,
Rui,Qiao, Zhi,Sobrino, José: Satellite remote sensing of surface urban heat islands: Progress,
challenges, and perspectives, Remote Sens., 11, 48, 2019. 10.3390/rs11010048

Shen, P., Zhao, S., Zhou, D., Lu, B., Han, Z., Ma, Y., Wang, Y., Zhang, C., Shi, C., Song, L.:
Surface and canopy urban heat island disparities across 2064 urban clusters in China, Science
of the Total Environment, 955, https://doi.org/10.1016/j.scitotenv.2024.177035, 2024.

**6. Line 189-196: The explanation here relies on visual interpretation of Figs. 3 and 4. I think this part can be removed as Fig.5 shows more reliable statistical analyses.**

_Response:_ We apologize for the unclear description. As you correctly pointed out, this part does rely too much on visual interpretation, especially the analysis of Fig. 4. The reason we introduced urban morphology in Section 3.1 was to conduct a preliminary analysis here, laying the groundwork for the in-depth analysis in the following sections. In response to your comment, we have reorganized the relevant content to reduce reliance on visual interpretation and better connect it with the more reliable statistical analyses in Fig. 5. The revised content is shown in Lines 218-229:

" Spatial analysis of daytime CUHII (Figure 3a) reveals that the Second Ring Road exhibits the highest CUHII values across all metrics: 0.27°C during NHW periods, 0.65°C during HW periods, and a difference of 0.38°C between the two. Analysis of urban configuration structures (Figure 4a) shows that the Second Ring has the highest proportion of dense buildings, and the compact layout leads to the accumulation of solar radiation heat in dense building clusters during the day, which is difficult to diffuse (Ge et al., 2016). This may be an important reason for the increase in daytime

CUHII during the HW periods. The nighttime CUHII differs (Figure 3b), with the Fourth Ring having the highest CUHII (1.80°C during NHW periods, 2.52°C during HW periods, and a difference of 0.72°C between the two). The Fourth Ring exhibits the highest proportion of high-rise buildings (Figure 4b). The concentrated emission of anthropogenic heat sources, such as air conditioners, in these high-rise zones (Yin & Zhao, 2024) could potentially contribute to the intensification of nighttime CUHII during heatwave events. Thus, urban morphology may be an important factor for the formation of diurnal patterns of CUHII. In the following sections, this study will conduct more reliable analyses using machine learning and numerical simulation methods."

**7. Fig.5: Are these results from XGBoost model? How is the model evaluated? For daytime results, the correlation value is small for all indicators except for BCR, which is only about 0.3; This seems to suggest that model performance is bad, or no single indicator is powerful enough to explain the CUHI. For nighttime results, many 3D indicators have coefficients very closed to SVF, and thus it is hard to argue that SVF is the dominant factor. The results can be changed with slight modifications of the data or training processes. Without rigorous model validation, the SHAP results in Fig.6 are less meaningful.**

_**Response:**_ We apologize for the unclear description. Figure 5 presents statistical results based on the linear Pearson correlation model, which was used to conduct a preliminary analysis of the relationship between urban morphological indicators and CUHII before machine learning analysis. We have supplemented this explanation in the figure caption and the main text. As you pointed out, there were inappropriate descriptions of SVF. We have revised the relevant content accordingly.

In addition, as you emphasized, rigorous model validation is crucial for subsequent SHAP analysis. Regarding the evaluation of the XGBoost model, we have provided a detailed response and supplementary information in Comment 3, and here we will give a brief summary. In this study, the coefficient of determination ($R^2$) and root mean square error (RMSE) were selected as evaluation metrics for simulation

accuracy. We have added content related to the performance of the XGBoost model in predicting CUHII: for the test dataset (Fig. S1), the R² values all exceed 0.45, while the RMSE values are all within 0.05. These results indicate that the XGBoost model can be regarded as a reliable tool for fitting the relationship between CUHII and urban morphology.

**8. Fig.8: the derivation and meaning of PDP plots shall be elaborated for general readers not familiar with this method. Current discussions related to Figure 8 are hard to understand.**

***Response:*** Thank you for your valuable suggestion. We apologize for the insufficient explanation of the Partial Dependence Plot (PDP) method, which may have caused difficulties in understanding. Explainable machine learning techniques can help understand the prediction process of "black-box models", as well as how the relationships between variables change within their value ranges (Bansal & Quan, 2024). Such post-hoc explanation techniques can probe into the model to reveal the relationships between variables. Partial Dependence Plot (PDP) is a commonly used technique that can present the marginal effects of independent variables (Friedman, 2001). The generated plots show partial dependence function values, which are the average marginal effects on the prediction results (Molnar, 2020). The partial dependence function is defined as follows:

$$\hat{f}_s(x_s) = E_{x_c}[\hat{f}(x_s, x_c)] = \int \hat{f}(x_s, x_c) dP(x_c) \quad (1)$$

Where $x_s$ is the target feature whose effects are to be studied, $x_c$ are other marginalized features, and $P$ represents the marginal probability density. The function $\hat{f}_s$ can be estimated using the Monte Carlo approximation method, with the formula as follows:

$$\hat{f}_s(x_s) = \frac{1}{n} \sum_{i=1}^{n} \hat{f}\left(x_S, x_C^{(i)}\right) \quad (2)$$

where $x_C^{(i)}$ denotes the value of $x_c$ in the dataset, and $n$ is the sample size.

Due to the model-agnostic nature of the above PDP specification, it can be applied to both traditional linear regression models and machine learning models such as XGBoost. For linear models, PDP can present marginal effects when other

independent variables take their mean values; for machine learning models, PDP can present the relationships between variables based on the tree structure of the model (Bansal & Quan, 2024).

To address this, we have supplemented the derivation and core meaning of PDP plots at the beginning of the discussion on Figure 8, aiming to help general readers grasp the method first. Additionally, we have adjusted the expression of results related to Figure 8 to enhance readability. The revised content is shown in Lines 287-291.


*Response:* Thank you for your inquiry. We apologize for not clarifying the specific method of SVF adjustment. To address this, we have supplemented details on the modification of the physical domain: SVF in different scenarios was adjusted only by changing building heights (without altering street width, building area, or other spatial parameters), and the adjustment was applied uniformly across the entire simulation domain. The revised description is as follows:

"This section selected a 500-meter radius area around Station 651061 on the North Fourth Ring Road as the simulation region, where the BCR was 0.225 and the SVF was 0.76. Three scenarios were set up by adjusting building heights (with street width, building footprint, and BCR kept unchanged to isolate the independent effect of SVF): ① Scenario I: Used the original building heights in the study area, corresponding to the real SVF (0.76, Figure 9a); ② Scenario II: Based on the PDP analysis results of the machine learning model, building heights were adjusted to reduce SVF to 0.735

(the critical point of positive/negative effects, Figure 9b); ③ Scenario III: Building heights were further adjusted to reduce SVF to 0.685 (the rapid growth stage of negative effects, Figure 9c). Notably, building height modifications were applied uniformly across the entire simulation domain to ensure consistent spatial conditions except for SVF differences."

**10. Figs. 10 and 11: After changing SVF in the ENVI-met domain, authors only analyze the temperature at the central point of the domain, this is too simple. In fact, using ENVI-MET at 1 neighborhood with different SVFs to demonstrate that temperature will change differently from NHW to HW does not sound convincing or necessary.**

*Response:* Thank you for your critical insight. We agree that analyzing only the central point temperature is insufficient to reflect spatial variations, and we apologize for the oversimplified interpretation. To address this, we have supplemented spatial heterogeneity analysis of temperature responses across the entire domain, rather than focusing solely on the central point. In addition, we have merged Figures 10 and 11 to facilitate the comparison of diurnal and nocturnal characteristics under different simulation scenarios. The revised content is as follows:

"The figure above shows the simulated AT spatial distribution under different scenarios during daytime (Figure 10a). Spatial patterns reveal that during NHW periods, Scenario II shows a 0.2–0.7°C temperature rise across the study region. The central point confirms this trend, with AT increasing from 30.68°C in Scenario I to 31.09°C in Scenario II. Meanwhile, Scenario III exhibits a 0.3–0.8°C cooling in these areas, driven by building shadows, with the central point AT in Scenario III decreasing to 30.33°C. During HW periods, these effects intensify. Scenario II sees a 0.5–1.1°C warming across these zones, with the central point air temperature in Scenario II increasing from 35.01°C to 35.76°C. Scenario III shows a 0.6–1.4°C cooling in study region, with the central point AT in Scenario III dropping to 34.39°C. As SVF decreased, the obstruction of building clusters to air flow intensified, reducing the heat dissipation capacity. Meanwhile, blocking of long wave radiation was

exacerbated, promoting heat accumulation and leading to temperature increases. It should be noted that the temperature change patterns in Scenario III, like the drop in central point AT, are related to excessively low SVF significantly increasing building shadow areas, enhancing the shading effect on solar radiation, thus reducing surface heat absorption and inhibiting temperature rise (Perini & Magliocco, 2014). Figure 10b shows the spatial distribution of the simulated AT indifferent scenarios at night. During NHW periods, the central point AT in Scenario I was 24.86℃, increasing to 25.10℃ in Scenario II with a relatively small variation, while that in Scenario III increased significantly to 25.90℃. During HW periods, the central point AT in Scenario I was 26.25℃, increasing to 26.83℃ in Scenario II and increased significantly to 27.93℃ in Scenario III. Notably, this pattern of temperature variation (moderate rise in Scenario II, sharp increase in Scenario III) is consistent across the entire simulation domain. The increase in building height hinders the convective heat dissipation of nighttime air, making heat dissipation difficult and thus promoting a significant temperature rise (Mo et al., 2024). Furthermore, the temperature differences between the scenarios during the HW periods were more significant than during the NHW periods, indicating that changes in building height have a more pronounced impact on air temperature during the HW periods, further amplifying the non-linear modulation of the building SVF in AT."

[Figure]

**Figure 10:** Spatial distributions of simulated AT across scenarios during daytime (a) and nighttime (b). NHW-SI represents Scenario I during NHW periods, HW-SI represents Scenario I during HW periods, and so forth. The intersection of the two gray crosshairs in each subplot indicates the location of the meteorological station.

11. Section 4: the discussion section focuses on analyzing the impact of wind on CUHI. However, the correlation is very weak. In addition, this part seems to deviate from previous correlation and SHAP results. From my perspective, authors seem to combine too many methods (XGBoost, ENVI-met, and correlation with wind) to explain CUHI change under HW, and this paper lacks a good organization and logic flow. After reading the paper, I am not sure what authors aim to address, and what are the key findings.

***Response:*** Thank you very much for your valuable comments, which have helped us identify critical issues in our discussion. We sincerely apologize for the unclear presentation in Section 4. We have carefully revised this section to address these concerns, and the key explanations are as follows:

"Figure 12a shows that during the daytime, the correlation coefficients (r) between WS and CUHII were -0.14 during NHW periods and -0.18 during HW periods, indicating a weak negative correlation that was slightly stronger during HW periods. Deng et al. (2025) simulated that a 10% increase in WS could reduce the CUHII by 0.16°C during summer days. Stronger solar radiation during HW periods makes the heat dissipation effect of wind more significant for CUHII. During night (Figure 12b), the r was -0.19 during NHW periods and -0.27 during HW periods, with enhanced negative correlations compared to daytime, especially during HW periods. This may be related to the heat dissipation characteristics of the underlying urban surface during nighttime (Liu et al., 2022), where slower heat release makes the modulation of WS in CUHII more pronounced. Notably, compared with research findings from other cities (Yang et al., 2023; Rajagopal et al., 2023; Deng et al., 2025), the CUHII in Beijing exhibits a unique characteristic—it is insensitive to WS variations both during the daytime and nighttime. This phenomenon may be explained by the regulatory role of local geographical environments: existing studies have confirmed that local circulations formed under different geographical backgrounds can significantly reshape the spatiotemporal distribution of urban extreme high temperatures (Zhang et al., 2011; Zhou et al., 2020; Chen et al., 2022). Specifically for Beijing, the mountainous terrain in its western and northern regions gives rise to a typical mountain-valley wind circulation, which exerts a strong regulatory effect on the urban near-surface thermal dynamic field (Miao et al., 2013). Observations show that wind directions in Beijing's urban area display a regular diurnal variation: northerly winds (mountain breeze) dominate from 05:00 to 10:00 Beijing Time; there is an obvious reversal around 11:00, shifting to southerly winds (valley breeze) which persist until 04:00 the next day. Additionally, the average wind speed of mountain breeze is significantly lower than that of valley breeze (Zheng et al., 2018b). Such distinct

periodic characteristics make mountain-valley breeze a key local factor influencing Beijing's thermal environment (Dou et al., 2014). Based on this, we speculate that the "insensitivity of CUHII to WS variations" observed in this study may be the result of interactions between the mountain-valley breeze cycle and the inherent diurnal cycle of CUHII—the superposition of these two periodic processes may weaken the regulatory effect of WS variations on CUHII. "

In short, the coupling mechanism between mountain-valley breezes and the diurnal cycle of CUHII (Fig. R1) may hold the key to explaining how WS acts on CUHII. We will further quantify this mechanism through refined numerical simulations in future research.

[Figure]

**Figure R1:** Schematic diagram illustrating the modulation of CUHII by mountain-valley breeze (self-draw).

In addition, We apologize for the unclear organization and vague presentation of the research objectives and key findings in the original manuscript, which may have caused confusion about how we integrated the methods (XGBoost, ENVI-met, and wind correlation analysis) and the core logic. To fix this, we have revised the abstract and conclusion. We now clarify that the methods work together (rather than being simply combined) and clearly present the central research objective and key findings. Specifically:

Revised abstract: Currently, the diurnal asymmetric and nonlinear mechanisms by which urban morphology modulates the canopy urban heat island (CUHI) during heat wave (HW) periods remain underexplored. This study aims to fill this gap by focusing on the area within the Fifth Ring Road of Beijing, integrating three complementary methods: XGBoost (to identify key morphological drivers), ENVI-met (to reveal

nonlinear regulatory processes), and wind environment analysis (to supplement dynamic modulation). The results show that: (1) HW periods significantly enhance CUHI intensity (CUHII) compared to non-heat wave (NHW) periods, with a 91.3% increase in daytime and 52.7% at night; (2) XGBoost identifies building coverage ratio (BCR) as the core daytime driver of CUHII, while sky view factor (SVF) dominates at night, and both 2D and 3D morphological indicators exert stronger effects during HW periods; (3) ENVI-met simulations reveal nonlinear mechanisms of building height/SVF: daytime thermal environments are co-driven by short-wave radiation shading and ventilation resistance (as SVF decreases), while nighttime environments are dominated by long-wave radiation accumulation by buildings; (4) Wind environment analysis further shows diurnal differences in wind's role: nighttime ventilation corridors mitigate CUHII by 33.91–42.09%, while daytime prevailing winds may exacerbate downstream CUHII via thermal advection. These findings clarify the diurnal asymmetric mechanisms of CUHI and provide scientific support for urban morphological optimization under extreme heat.

Revised conclusions: By integrating ground observations, XGBoost, and ENVI-met simulations, this study systematically unravels the diurnal asymmetric and nonlinear response of canopy urban heat island (CUHI) to urban morphology during heat wave (HW) periods in Beijing. The results show that compared with non-heat wave (NHW) periods, CUHI intensity (CUHII) during HW periods is significantly enhanced, with a 91.3% increase in daytime and 52.7% at night, and its diurnal variation presents a U-shaped fluctuation with distinct spatial patterns (strongest within the Second Ring Road in daytime and most prominent around the Fourth Ring Road at night). Machine learning analysis indicates that building coverage ratio (BCR) is the most critical driver of daytime CUHII, while sky view factor (SVF) dominates at night; the mean importance of 2D/3D morphological indicators increases by 16.2%–36.7% during HW periods, with significant interactions between BCR and SVF. ENVI-met simulations further confirm the nonlinear modulation mechanism of urban morphology: when SVF decreases from 0.735 to 0.685, daytime temperature regulation is jointly affected by short-wave radiation shading and ventilation

resistance, showing a "first warming then cooling" pattern, while nighttime temperature changes are dominated by the reflection and accumulation of long-wave radiation by buildings, exhibiting accelerated warming characteristics. Additionally, the study identifies diurnal differences in the impact of wind fields on CUHII: ventilation corridors can reduce nighttime CUHII by 33.91%–42.09% to mitigate heat islands effectively, whereas daytime prevailing winds may intensify CUHII in downstream regions through thermal advection rather than simply acting as a cooling factor. These findings clarify the diurnal asymmetric formation mechanism of CUHI during HW periods and provide quantitative references for optimizing urban morphology and planning ventilation corridors, offering precise scientific guidance for mitigating urban thermal risks.

**Minor comments:**

**1. Fig.1 caption: "Overview of study area" is repeated.**

*Response:* Thank you for your comment. We apologize for the repetition of "Overview of study area" in the caption of Fig. 1. This issue has been corrected. Additionally, we have carefully checked the entire manuscript to avoid similar writing problems.

**2. Line 167: remove "the next day" as this is a averaged diurnal cycle**

*Response:* Thank you for your reminder. We have removed "the next day" from Line 167, as it is indeed inappropriate in the context of an averaged diurnal cycle.

**3. Fig.3 caption: only (a) and (b) sub-figures; and I suggest authors to add the different in CUHI between HW and NHW to better illustrate the distribution of CUHI change**

*Response:* Thank you for your professional suggestion. As you recommended, we have added the difference in CUHI between HW and NHW to Fig. 3 and conducted relevant analyses in the text.

[Figure]

**Figure: 3** Diurnal spatial patterns of CUHII during NHW & HW. Panel (a) for daytime, (b) for nighttime. In each panel, left: NHW CUHII stats & distribution; middle: HW CUHII stats & distribution; right: HW-NHW CUHII difference.

---

## Author Comment (AC2)

**Response to Review Comments of the Second Reviewer**

Dear Reviewer and Editors:

We are sincerely grateful to the editor and reviewer for their valuable time for reviewing our manuscript. The comments are very helpful and valuable, and we have addressed the issues raised by the reviewer in the revised manuscript. Please find our point-by-point response (in blue text) to the comments (in black text) raised by the reviewer. We have revised the paper according to your comments (highlighted in red text of the revised manuscript).

Sincerely yours,

Dr. Yuanjian Yang, representing all co-authors

**Major comments:**

**1. Section 2.3.1: The HW definition requires stronger justification. Specifically, why did the authors decide to use reference stations to define HW and why was the threshold set to "more than two reference stations"?**

*Response:* We apologize for the unclear in the original manuscript regarding the HW events definition. We have supplemented relevant details in the revised version to address this issue.

Reference stations (primarily rural stations) provide a baseline of regional climatic conditions unaffected by urbanization, ensuring that the defined HWs reflect true regional extreme HW events rather than local CUHI effects. As highlighted in previous studies (Cheng et al., 2020; Stewart & Oke, 2012), rural reference stations, with minimal impervious surfaces and anthropogenic heat emissions, capture the natural climatic background.

Heat waves, by nature, are large-scale extreme weather events (Perkins et al., 2012;

Rajulapati et al., 2022), and a single reference station's abnormal high temperatures may result from local factors (e.g., microtopography, temporary human activities) rather than a true regional HW. Requiring confirmation from multiple reference stations reduces the risk of misclassification due to individual station errors or local anomalies, improving the robustness of the definition. This aligns with the statistical logic in our study, where HW were counted independently at each station but required spatial consistency to be recognized as a regional event (Xue et al., 2023).

In summary, using reference stations ensures the HW definition is rooted in regional climatic anomalies, while the multi-station threshold guarantees the spatial generality of the identified heat waves, making the results more reliable for analyzing CUHI and HW.

**References:**

Cheng, X., Lan, T., Mao, R., Gong, D., Han, H., Liu, X.: Reducing air pollution increases the local diurnal temperature range: a case study of Lanzhou, China, Meteorological Applications, 27, https://doi.org/10.1002/met.1939, 2020.

Perkins, S. E., Alexander, L. V., & Nairn, J. R.: Increasing frequency, intensity and duration of observed global heatwaves and warm spells, Geophysical Research Letters, 39, https://doi.org/10.1029/2012GL053361, 2012.

Rajulapati, C. R., Gaddam, R. K., Nerantzaki, S. D., Papalexiou, S. M., Cannon, A. J., Clark, M. P.: Exacerbated heat in large Canadian cities, Urban Climate, 42, 101097, https://doi.org/10.1016/j.uclim.2022.101097, 2022.

Stewart, I. D., & Oke, T. R.: Local climate zones for urban temperature studies, Bulletin of the American Meteorological Society, 93, 1879–1900, https://doi.org/10.1175/BAMS-D-11-00019.1, 2012.

Xue, J., Zong, L., Yang, Y., Bi, X., Zhang, Y., Zhao, M.: Diurnal and interannual variations of canopy urban heat island (CUHI) effects over a mountain–valley city with a semi-arid climate, Urban Climate, 48, 101425, https://doi.org/10.1016/j.uclim.2023.101425, 2023.

**2. Section 2.3.3: More details on the training/validation processes of XGBoost are needed. How were the collinearity among morphological indicators (e.g.,**

**FAR and BCR) treated in the ML models? More detailed explanations of SHAP and PDP methods would improve reader comprehension of the results in Figure 6-8.**

*Response:* Thank you for your constructive comments. We appreciate the opportunity to clarify the methodological details, and we have supplemented Section 2.3.3 with additional explanations as follows:

(1) Details on the training/validation processes of XGBoost:

The XGBoost model training and validation processes were designed to ensure robustness:

"In this study, we first performed iterative calculations on 7 commonly used hyperparameters (eta, gamma, max_depth, min_child_weight, subsample, colsample_bytree, and nrounds) within a preset hyperparameter tuning space, and selected the optimal hyperparameter combination that minimizes model error using a 5-fold cross-validation method (Yang et al., 2020; Lin et al., 2024). After completing hyperparameter optimization, we randomly split the sample points in the Yangtze River Basin at a 7:3 ratio to obtain training samples (70%) and validation samples (30%), which were used for training and validating the XGBoost model, respectively. Meanwhile, the coefficient of determination ($R^2$) and root mean square error (RMSE) were chosen as evaluation metrics for simulation accuracy. "


To address collinearity among morphological indicators in the XGBoost model, we first conducted a correlation analysis for feature screening, following established methodologies in similar studies (Harrell, 2015). Specifically, pairwise Pearson

correlation coefficients were calculated for all indicators, with a threshold of 0.8 set to identify highly collinear features. Features exceeding this threshold were evaluated for retention based on their physical significance and relevance to CUHII. For FAR (Floor Area Ratio) and BCR (Building Coverage Ratio), their correlation coefficient was 0.56, well below the 0.8 threshold, indicating moderate correlation without severe collinearity. Thus, both indicators were retained in the model.

Notably, the correlation coefficient between H (average building height) and H-std (building height standard deviation) exceeded 0.8. Both are critical urban morphological parameters influencing the local thermal environment (Tian et al., 2019), yet their regulatory mechanisms differ substantially. On one hand, taller buildings reduce daytime surface temperatures by increasing shading and reducing solar radiation input at the surface (Zhang et al., 2016; Krayenhoff & Voogt, 2016; Taleghani et al., 2016; Cai, 2017). Conversely, high-rise buildings have higher heat capacity, enabling heat storage and slow nighttime release, which delays cooling and intensifies nocturnal heat island effects (Unger, 2004). Additionally, ventilation resistance increases with H: taller buildings strongly block air flow, potentially causing stagnant air in the urban canopy and localized heat accumulation (Hang et al., 2011). In contrast, H-std (building height standard deviation) captures height variation, reflecting spatial heterogeneity of urban morphology with distinct thermal regulatory roles. By day, greater H-std enhances urban canopy turbulence, promoting air circulation and heat dissipation to reduce LST. For instance, studies in Shenzhen show significant cooling when ln(H-std) > 0.5, as increased height variation strengthens surface roughness and airflow disturbance (Wan et al., 2025). Similarly, Fuzhou's BH_std (building height standard deviation) correlates negatively with daytime LST, indicating ventilation-driven cooling (Lin et al., 2024). Given their distinct roles in regulating thermal environments, H and H-std are irreplaceable. Therefore, this study retained both indicators.


In addition, Partial Dependence Plot (PDP) is a commonly used technique that can present the marginal effects of independent variables (Friedman, 2001). The generated plots show partial dependence function values, which are the average marginal effects on the prediction results. The partial dependence function is defined as follows:

$$\hat{f}_s(x_s) = E_{x_c}[\hat{f}(x_s, x_c)] = \int \hat{f}(x_s, x_c) dP(x_c) \quad (1)$$

Where $x_s$ is the target feature whose effects are to be studied, $x_c$ are other marginalized features, and $P$ represents the marginal probability density. The function $\hat{f}_s$ can be estimated using the Monte Carlo approximation method, with the formula as follows:

$$\hat{f}_s(x_s) = \frac{1}{n} \sum_{i=1}^{n} \hat{f}\left(x_S, x_C^{(i)}\right) \quad (2)$$

where $x_C^{(i)}$ denotes the value of $x_c$ in the dataset, and $n$ is the sample size.

Due to the model-agnostic nature of the above PDP specification, it can be applied to both traditional linear regression models and machine learning models such as XGBoost. For linear models, PDP can present marginal effects when other independent variables take their mean values; for machine learning models, PDP can present the relationships between variables based on the tree structure of the model (Bansal & Quan, 2024).

To address this, we have supplemented the derivation and core meaning of PDP to enhance clarity:

"Partial dependency plots (PDP) are a common explainable machine learning technique that reveals the marginal effect of a target feature (e.g., urban

morphological indicators) on prediction outcomes (CUHII) by holding other features at their average levels or marginalizing their effects (Friedman, 2001; Bansal & Quan, 2024). Specifically, PDP illustrates the average trend of change in CUHII as a single indicator (or a combination of two indicators) varies, while other indicators remain stable—thereby isolating the independent impact of the target indicator. By leveraging PDP to visualize the functional relationship between feature variables and model outputs, we clarify the marginal effects of urban morphological indicators on CUHII, which supports the identification of key driving factors and their threshold characteristics. "

These supplementary ensure the methodological rigor of the ML-based analyses and improve the interpretability of results in Figures 6–8. We appreciate your guidance in strengthening the methodological transparency of our study.

To address this, we have added a new figure (now Figure X, placed at the beginning of Section 3.2) that demonstrates the model's performance. This figure includes: (1) a

scatter plot of observed vs. predicted CUHII values, with a fitted regression line; (2) statistical metrics such as R², RMSE, and MAE to quantify prediction accuracy. These results confirm that the XGBoost model achieves robust predictability, providing a reliable basis for the subsequent SHAP and partial dependency analyses.

We appreciate your insight, which has enhanced the rigor of our methodological validation.

[Figure]

**Figure S1:** The performance graph of the XGBoost model in predicting CUHII.

**4. Line 255-257: This summary largely repeats content in line 236-239. The authors should streamline the conclusion from figure 8, e.g., focus more on the nonlinear modulation.**

*__Response:__* We apologize for the redundant in the previous version. We have deleted the content in lines 236–239 to avoid repetition. Based on the findings from Figure 8, we have streamlined the conclusion regarding nonlinear modulation. The revised description is as follows:

"In summary, the regulation of urban morphology on CUHII exhibits significant

diurnal asymmetry: 2D indicators predominate during the daytime, while 3D indicators play a dominant role at night. Furthermore, urban morphology exerts nonlinear modulation on CUHII, characterized by threshold effects and dual roles (e.g., SVF showing both negative and positive impacts), with these nonlinear effects being more pronounced during HW periods."

**5. Section 3.3: The scenario setup requires clarification. How were the uniform SVF values applied across the entire domain in scenario II and III? Does scenario I have spatially heterogeneous SVF values? If so, the rationale for using uniform values in scenario II and III needs justification. Currently it is difficult to interpret spatial changes in Figures 10-13 with most discussions focused on the central point.**

*Response:* Thank you for your valuable comment. We apologize for the insufficient clarification on scenario setup and spatial characteristics of SVF, which has led to difficulties in interpretation. We have revised the relevant content to address these concerns, and the key explanations are as follows:

(1) SVF characteristics in Scenario I: Scenario I adopted the original building heights of the study area (a 500-meter radius around Station 651061), where building heights vary spatially due to real urban morphological heterogeneity (e.g., differences in residential and commercial building heights). Consequently, the SVF in Scenario I is spatially heterogeneous, with local variations around the mean value of 0.76, reflecting the actual urban spatial pattern.

(2) Uniform adjustment of SVF in Scenarios II and III: To isolate the independent effect of SVF on thermal environment, we adjusted building heights uniformly across the entire domain in Scenarios II and III (while keeping street width, building footprint, and BCR unchanged). This uniform adjustment ensured that SVF values in these scenarios are more spatially consistent (targeting 0.735 and 0.685, respectively), reducing interference from other spatial heterogeneities (e.g., uneven building height distribution). This design allows us to explicitly link temperature changes to SVF variations, avoiding confounding effects from concurrent changes in multiple

morphological factors.

(3) Enhanced spatial analysis: We agree that analyzing only the central point temperature is insufficient to reflect spatial variations, and we apologize for the oversimplified interpretation. To address this, we have supplemented spatial heterogeneity analysis of temperature responses across the entire domain, rather than focusing solely on the central point. In addition, we have merged Figures 10 and 11, Figures 12 and 13 to facilitate the comparison of diurnal and nocturnal characteristics under different simulation scenarios. The revised content is as follows:

[revised manuscript text omitted]

**6. Section 4: While the wind-CUHII relationship is worth discussing, the analysis should emphasize how urban morphology modulates wind patterns to be tightly connected with the main theme of this work. The current presentation of Figures 14-16 lacks clear connection to morphological controls, making it difficult to identify the key messages.**

*Response:* Thank you very much for your valuable comments, which have helped us identify critical issues in our discussion. We sincerely apologize for the unclear presentation in Section 4. We have carefully revised this section to address these concerns, and the key explanations are as follows:

"Figure 12a shows that during the daytime, the correlation coefficients (r) between WS and CUHII were -0.14 during NHW periods and -0.18 during HW periods, indicating a weak negative correlation that was slightly stronger during HW periods. Deng et al. (2025) simulated that a 10% increase in WS could reduce the CUHII by 0.16°C during summer days. Stronger solar radiation during HW periods makes the heat dissipation effect of wind more significant for CUHII. During the night (Figure 12b), the r was -0.19 during NHW periods and -0.27 during HW periods, with enhanced negative correlations compared to daytime, especially during HW periods. This may be related to the heat dissipation characteristics of the underlying urban surface during nighttime (Liu et al., 2022), where slower heat release makes the modulation of WS in CUHII more pronounced. Notably, compared with research findings from other cities (Yang et al., 2023; Rajagopal et al., 2023; Deng et al., 2025), the CUHII in Beijing exhibits a unique characteristic—it is insensitive to WS variations both during the daytime and nighttime. This phenomenon may be closely linked to urban morphology and local geographical environments. Urban morphology significantly modulates wind penetration and heat exchange efficiency: compact built-up areas with high BCR and low SVF (e.g., the Second Ring Road) form dense building clusters that block airflow, reducing WS and weakening wind-driven heat dissipation, thus making CUHII less responsive to WS changes. In addition, existing studies have confirmed that local circulations formed under different geographical backgrounds can significantly reshape the spatiotemporal distribution of urban extreme high temperatures (Zhang et al., 2011; Zhou et al., 2020; Chen et al., 2022). Specifically for Beijing, the mountainous terrain in its western and northern regions gives rise to a typical mountain-valley wind circulation, which interacts with urban morphology: dense buildings in central areas disrupt valley breeze penetration, while sparse layouts in suburbs align with mountain winds. This interplay between morphology and terrain-induced winds weakens the modulation of WS variations on CUHII. Observations show that wind directions in Beijing's urban area display a regular diurnal variation: northerly winds (mountain breeze) dominate from 05:00 to 10:00 Beijing Time; there is an obvious reversal around 11:00, shifting to southerly

winds (valley breeze) which persist until 04:00 the next day. Additionally, the average WS of the mountain breeze is significantly lower than that of the valley breeze (Zheng et al., 2018). Such distinct periodic characteristics make mountain-valley breeze a key local factor influencing Beijing's thermal environment (Dou et al., 2014). Based on this, we speculate that the "insensitivity of CUHII to WS variations" observed in this study may be the result of interactions between the mountain-valley breeze cycle, urban morphology, and the inherent diurnal cycle of CUHII."

In short, the coupling mechanism between local circulation, urban morphology and the diurnal cycle of CUHII (Fig. R1) may hold the key to explaining how WS acts on CUHII. We will further quantify this mechanism through refined numerical simulations in future research.

[Figure]

**Figure R1:** Schematic diagram illustrating the modulation of CUHII by mountain-valley breeze (self-draw).

[revised manuscript text omitted]

**Minor comments:**

**1. Figure 3: There are (a)-(f) in the caption but only four subplots are presented.**

***Response:*** We apologize for the error in Figure 3 where the caption incorrectly. This has been corrected to ensure the number of subplots matches the caption. We have also thoroughly reviewed the entire manuscript to prevent similar issues in other figures. Thank you for bringing this to our attention.

[Figure]

**Figure: 3** Diurnal spatial patterns of CUHII during NHW & HW. Panel (a) for daytime, (b) for nighttime. In each panel, left: NHW CUHII stats & distribution; middle: HW CUHII stats & distribution; right: HW-NHW CUHII difference.

**2. Figure 6: It would be easier to interpret if the authors can group 2D and 3D indicators (e.g.presenting all 2D indicators in the first 6 rows, followed by 3D indicators).**

*Response:* We apologize for the unclear differentiation of 2D and 3D urban morphological indicators in Figure 6, which compromised interpretability. The current arrangement of indicators in the SHAP plots follows the feature importance order automatically determined by the XGBoost model (based on their contribution to CUHII predictions), resulting in the interleaving of 2D (e.g., BCR, L/W) and 3D (e.g., H, SVF, H-std) indicators.

Recognizing the merit of your suggestion, we have enhanced the figure by color-coding 2D indicators in red and 3D indicators in green. Thank you for your constructive feedback, which has guided us to strengthen the figure's communicative clarity.

[Figure]

**Figure 6:** SHAP value analysis of urban morphology indicators for diurnal CUHII during NHW and HW periods, using XGBoost model. SHAP quantifies feature contributions to model outputs. The red/blue color gradients represent high/low feature values, with red indicating 2D urban morphological indicators and green indicating 3D

urban morphological indicators.

**3. Figure 9d: Does the 'simulation accuracy' refer to scenario I?**

*Response:* We apologize for the ambiguous reference to "simulation accuracy" in Figure 9d. To address this, we have explicitly clarified in the figure caption. The updated caption now states: "(d) simulation accuracy of air temperature (AT) for Scenario I during NHW and HW periods."

---

## Author Comment (AC3)

**Response to Review Comments of the Reviewer**

Dear Reviewer and Editors:

We are sincerely grateful to the editor and reviewer for their valuable time for reviewing our manuscript. The comments are very helpful and valuable, and we have addressed the issues raised by the reviewer in the revised manuscript. Please find our point-by-point response (in blue text) to the comments (in black text) raised by the reviewer. We have revised the paper according to your comments (highlighted in red text of the revised manuscript).

Sincerely yours,

Dr. Yuanjian Yang, representing all co-authors

**I would like to thank the authors for their efforts in addressing my comments posted in the first round. Most of my concerns are solved except for the performance of XGBoost.**

*Response:* Thank you for your recognition of our revised work. We apologize for the unclear description of the XGBoost model's performance in the previous version. We have provided a detailed response to this concern in the following section, and we hope this addresses your remaining worry.

**1) Regarding my comments on the performance of XGBoost, authors add Figure S1to show that RMSE values are smaller than 0.05 (what is the unit) with R2 values around 0.5-0.6. How could the RMSE values be so small while the R2 is not high? This is crtical as XGBoost is used in subsequent analyses.**

*Response:* Thank you very much for your careful review and professional comments. We made a critical error during modeling: the target variable (CUHII) was mistakenly reversed with the feature variables (urban form indicators). This led to the incorrect

compression of the predicted value range of CUHII in the original Figure S1 (e.g., the CUHII in the first panel was only 0~0.5°C, which contradicts its practical physical meaning).

After correcting the variable correspondence, the new results (second figure) align with the true amplitude characteristics of CUHII: during the daytime, CUHII is generally low (most actual values are < 2.0°C), so the RMSE is small (0.208~0.254°C); at night, especially during heat wave (HW) periods, CUHII itself has a larger amplitude (maximum value exceeding 3.0°C). Even though the $R^2$ for this period is relatively higher (0.465), the absolute error (RMSE = 0.663°C) has increased significantly due to the expanded amplitude—which is consistent with the rule that "the larger the variable amplitude, the naturally wider the range of absolute error".

In addition, we have reviewed all figures and tables in the manuscript one by one, corrected details such as data mapping and variable labeling, and prevented similar logical errors from occurring again. Thank you for helping us correct this core issue and ensuring the reliability of the research conclusions.

[Figure]

**Figure. S1** The performance graph of the XGBoost model in predicting CUHII.